# A Computationally Efficient Sparsified Online Newton Method

## Abstract

Second-order methods have enormous potential in improving the convergence of deep neural network (DNN) training, but are prohibitive due to their large memory and compute requirements. Furthermore, computing the matrix inverse or the Newton direction, which is needed in second-order methods, requires high precision computation for stable training as the preconditioner could have a large condition number. This paper provides a first attempt at developing computationally efficient sparse preconditioners for DNN training which can also tolerate low precision computation. Our new Sparsified Online Newton (SONew) algorithm emerges from the novel use of the LogDet matrix divergence measure; we combine it with sparsity constraints to minimize regret in the online convex optimization framework. Our mathematical analysis allows us to reduce the condition number of our sparse preconditioning matrix, thus improving the stability of training with low precision. We conduct experiments on a feed-forward neural-network autoencoder benchmark, where we compare training loss of optimizers when run for a fixed number of epochs. In the float32 experiments, our methods outperform the best-performing first-order optimizers and perform comparably to Shampoo, a state-of-the-art second-order optimizer. However, our method is even more effective in low precision, where SONew finishes training considerably faster while performing comparably with Shampoo on training loss.

## 1 Introduction

Stochastic first order methods which use the negative gradient direction to update parameters have become the standard for training deep neural networks (DNNs). Gradient-based preconditioning involves finding an update direction, by multiplying the gradient with a preconditioner matrix carefully chosen from gradients observed in previous iterations, to improve convergence. (Full-matrix) Adagrad (Duchi et al., 2011b), online Newton method (Hazan et al., 2007) and natural gradient descent (Amari, 1998) use a full-matrix preconditioner, but computing and storing the full matrix is infeasible when there are millions of parameters. Thus, diagonal versions such as diagonal Adagrad, Adam (Kingma & Ba, 2014), and RMSprop (Hinton et al., 2012) are now widely used to train DNNs due to their scalability.

Several higher-order methods have previously been applied to deep learning (Gupta et al., 2018; Anil et al., 2020; Goldfarb et al., 2020; Martens & Grosse, 2015). All these methods use Kronecker products that reduce computational and storage costs to make them feasible for training neural networks. However, these methods rely on matrix inverses or pth-roots that require high precision arithmetic as the matrices they deal with can have large condition numbers (Anil et al., 2020; 2022). Meanwhile, deep learning hardware accelerators have evolved towards using lower precision (bfloat16, float16, int8) (Henry et al., 2019; Jouppi et al., 2017) to reduce overall computational and memory costs and improve training performance. This calls for further research in developing efficient optimization techniques that work with low precision. Indeed there is recent work along these directions, from careful quantization of Adam (Dettmers et al., 2021) to 8-bits to optimizer agnostic local loss optimization (Amid et al., 2022) that leverage first-order methods to match higher-order methods.

In this paper, we present a first attempt towards computationally efficient *sparse* preconditioners for DNN training. Regret analysis when using a preconditioner reveals that the error is bounded by two summations (see (3) below); the first summation depends on the change in the preconditioning matrix, while the second depends on the generalized gradient norm. We take the approach

of minimizing the second term while regularizing two successive preconditioners to be close in the LogDet matrix divergence measure (Kulis et al., 2009). This technique gives us an Online Newton method (Hazan et al., 2007). To make it computationally efficient, we further sparsify the preconditioner by finding a sparse approximation that is close in LogDet divergence. Thus we are consistent in using the same measure (LogDet divergence) in both the regularization and sparsification steps. This gives us our Sparsified Online Newton (SONew) method, which only requires $\mathcal{O}(n)$ time and memory complexity per iteration. We achieve this by imposing structured sparsity, such as tridiagonal and banded sparsity patterns, in the preconditioner. This is unlike most existing online Newton methods that require at least $\mathcal{O}(n^2)$ space and time complexity. By making each step linear time, the SONew method can be applied to train modern DNNs. Further, for some sparsity structures, our method is easily parallelized thus making negligible the overhead of computing the preconditioner. We also show that introducing sparsity allows us to reduce the condition number of the problem; as a consequence our preconditioner allows us to train DNNs even in low precision arithmetic.

We establish regret bound guarantees of our algorithm in the online convex optimization framework. This involves using various properties about LogDet divergence and connections to other Bregman matrix divergences (Bregman, 1967), such as the von Neumann matrix divergence (Kulis et al., 2009). We conduct experiments on an MLP Autoencoder, where we obtain better training loss compared to first order methods. We also conduct experiments on large-scale benchmarks in Appendix A.5, and observe comparable or improved performance than Adam (Kingma & Ba, 2014). Our MLP experiments in limited precision arithmetic (bfloat16) showed comparable performance with second-order methods, while being considerably faster.

## 2 BACKGROUND

The inner product between matrices is defined as $\langle A, B \rangle = \mathrm{Tr}(A^T B)$, where $\mathrm{Tr}(.)$ denotes the matrix trace. The Frobenius norm of a matrix $A$ is $\|A\|_F = \sqrt{\mathrm{Tr}(A^T A)}$, while its spectral norm is $\|A\|_2 = \max_x \|Ax\|_2 / \|x\|_2$. We use $I_n \in \mathbb{R}^{n \times n}$ to denote an identity matrix. We use $S_n, S_n^{++}$ to denote the set of symmetric, and positive definite matrices respectively. The generalized norm of a vector $x \in \mathbb{R}^n$ with respect to matrix $A \in S_n^{++}$ is defined as $\|x\|_A = \sqrt{x^T A x}$. We use $\det(A)$ to denote the determinant of matrix $A$, and $\mathrm{diag}(A)$ to denote diagonal matrix with $\mathrm{diag}(A)_{ii} = A_{ii}$. We use $\mathcal{G}$ and $\tilde{\mathcal{G}}$ to denote a graph and its sub-graph with a vertex set $[n] = \{1, \ldots, n\}$. Let $E_\mathcal{G}$ denote set of edges in graph $\mathcal{G}$, and $\mathrm{neig}_\mathcal{G}(i)$ denote neighbours of vertex $i$ in graph $\mathcal{G}$. The sparsity graph/pattern of a matrix $A \in \mathbb{R}^{n \times n}$ is a graph $\mathcal{G}$ with edges $E_\mathcal{G} = \{(i, j) : A_{ij} \neq 0\}$ corresponding to the non-zero entries in $A$. We extensively use $S_n(\mathcal{G})^{++}$ to denote the set of positive definite matrices with sparsity structure given by graph $\mathcal{G}$. Given an index set $I = \{i_1, i_2, .., i_n\}$, we use $A_{II}$ to denote the corresponding principal sub-matrix of $A$.

### 2.1 LOGDET MATRIX DIVERGENCE

Let $\phi : S_n^{++} \to \mathbb{R}$ be a strictly convex, differentiable function. Then the Bregman matrix divergence between $X, Y \in S_n^{++}$ is defined as (Bregman, 1967; Kulis et al., 2009):

$$D_\phi(X, Y) = \phi(X) - \phi(Y) - \mathrm{Tr}(\nabla\phi(Y)^T(X - Y)).$$

Since $\phi$ is convex, $D_\phi(X, U) \geq 0$ for all $X, Y \succ 0$. A well known example is when $\phi(X) = \|X\|_F^2$, the corresponding Bregman divergence $D_\phi(X, Y) = \|X - Y\|_F^2$ is the squared Frobenius norm. In this paper, we extensively use the divergence when the convex function is $\phi(X) = -\log\det(X)$; the corresponding divergence measure $D_{\ell d}(X, Y)$ is called the *LogDet matrix divergence*:

$$D_{\ell d}(X, Y) = -\log\det(XY^{-1}) + \mathrm{Tr}(XY^{-1}) - n. \tag{1}$$

The LogDet divergence is scale invariant to invertible matrices $A$, i.e. $D_{\ell d}(A^T X A, A^T Y A) = D_{\ell d}(X, Y)$. The following revealing form of LogDet divergence in terms of eigendecompositions of $X = V\Lambda V^T$ and $Y = U\Theta U^T$ (Kulis et al., 2009):

$$D_{\ell d}(X, Y) = \sum_i \sum_j (v_i^T u_j)^2(\lambda_i/\theta_j - \log(\lambda_i/\theta_j) - 1). \tag{2}$$

These two properties are later used in Section 3 to highlight the significance of LogDet divergence in our algorithm.

## 3 SONew: Sparsified Online Newton Method

We now present our proposed SONew algorithm.

### 3.1 Regret minimization via LogDet divergence

We set up our problem under the online convex optimization framework (OCO) (Shalev-Shwartz et al., 2012; Hazan et al., 2016), where at each round the learner makes a prediction $w_t$ in an online fashion and receives a convex loss $f_t(w_t)$ and gradient $g_t = \nabla f_t(w_t)$ as feedback. The goal of the learner is to reduce regret $R_T$ by predicting $w_t$ so that a low aggregate loss $\sum_{t=1}^{T} f_t(w_t)$ is achieved compared to the best possible, $w^* = \arg\min_w \sum_{t=1}^{T} f_t(w)$. Formally, the regret is given by

$$R_T(w_1, \ldots, w_T) = \sum_{t=1}^{T} f_t(w_t) - \sum_{t=1}^{T} f_t(w^*).$$

To upper bound this regret, we proceed as in Hazan et al. (2016) by analyzing the error in the iterates for the update $w_{t+1} := w_t - \eta X_t g_t$, where $X_t \in \mathbb{R}^{n \times n}$. Then $\|w_{t+1} - w^*\|_{X_t^{-1}}^2 = \|w_t - \eta X_t g_t - w^*\|_{X_t^{-1}}^2 = \|w_t - w^*\|_{X_t^{-1}}^2 + \eta^2 g_t^T X_t g_t - 2\eta(w_t - w^*)^T g_t$. The convexity of $f_t$ implies that $f_t(w_t) - f_t(w^*) \le (w_t - w^*)^T g_t$ leading to $f_t(w_t) - f_t(w^*) \le \|w_t - w^*\|_{X_t^{-1}}^2 - \|w_{t+1} - w^*\|_{X_t^{-1}}^2 + \eta^2 g_t^T X_t g_t$. Summing over all $t \in [T]$ and rearranging reveals the following upper bound on overall regret:

$$R_T \le \frac{1}{2\eta} \|w_1 - w^*\|_{X_1^{-1}}^2 + \frac{1}{2\eta} \sum_{t=2}^{T} (w_t - w^*)^T (X_t^{-1} - X_{t-1}^{-1})(w_t - w^*) + \frac{\eta}{2} \sum_{t=1}^{T} g_t^T X_t g_t. \quad (3)$$

Since $w^*$ is unknown, finding $X_t$ which minimizes (3) is infeasible. So to minimize regret, we attempt to minimize the last term in (3) while regularizing $X_t^{-1}$ to be "close" to $X_{t-1}^{-1}$. The nearness measure we choose is the LogDet matrix divergence, thus leading to the following objective

$$X_t = \arg\min_{X \in S_n^{++}} g_t^T X g_t, \quad \text{such that} \quad D_{\ell d}(X, X_{t-1}) \le c_t, \quad (4)$$

where $D_{\ell d}$ is as in (1). Why do we use the LogDet divergence? From (2), due to the term $\lambda_i / \theta_j$, $D_{\ell d}(X, X_{t-1})$ prioritizes matching the smaller eigenvalues of $X_{t-1}$ with those of $X$, i.e., matching the larger eigenvalues of $X_{t-1}^{-1}$ and $X^{-1}$. As a consequence, LogDet divergence regularizes $X$ by matching up its large eigenvalues with those of $X_{t-1}$. For e.g., if smallest and largest eigenvalue of $X_{t-1}$ are $\theta_n$ and $\theta_1$, then for an eigenvalue $\lambda$ of $X$, when $\lambda > \theta_n, \theta_1$, the penalty from (2) for $\theta_n$ is higher than for $\theta_1$, $(\lambda/\theta_n - \log(\lambda/\theta_n) - 1) > (\lambda/\theta_1 - \log(\lambda/\theta_1) - 1)$. This intuition leads us to formulate (4) as our objective. We recall that there is precedence of using the LogDet divergence in the optimization literature; indeed the celebrated BFGS algorithm (Broyden, 1967; Fletcher, 1970; Goldfarb, 1970; Shanno, 1970) can be shown to be the unique solution obtained when the LogDet divergence between successive preconditioners, subject to a secant constraint, is minimized (as shown in the beautiful 4-page paper by Fletcher (1991)).

The optimization problem in (4) is convex in $X$ since the LogDet divergence is convex in its first argument. The Lagrange $\mathcal{L}(X, \lambda_t) = g_t^T X g_t + \lambda_t(D_{\ell d}(X, X_{t-1}) - c_t) = \text{Tr}(X g_t g_t^T) + \lambda_t(-\log\det(X X_{t-1}^{-1}) + \text{Tr}(X X_{t-1}^{-1}) - n))$. Setting $\nabla \mathcal{L}(X, \lambda_t) = 0$, and using the fact that $\nabla \log\det(X) = X^{-1}$ we get the following update rule:

$$X_t^{-1} = X_{t-1}^{-1} + g_t g_t^T / \lambda_t. \quad (5)$$

Note that setting $c_t = 0$ (equivalently $\lambda_t = \infty$) $\forall t \in [n]$ in (4) results in no change to the pre-conditioner in any round. In this case, with $X_0 = I_n$, we get online gradient descent (Zinkevich, 2003). On the other hand, setting $\lambda_t = 1$ gives the update rule of the online Newton method Hazan et al. (2007). Our update rule differs from (full-matrix) Adagrad (Duchi et al., 2011b) which has $X_t^{-2} = X_{t-1}^{-2} + g_t g_t^T$.

Maintaining and updating $X_t$ as in (5) is possible by using Sherman-Morrison formula but requires $\mathcal{O}(n^2)$ storage and time complexity. This becomes impractical when $n$ is in the order of millions which is typically the case in DNNs.

**Algorithm 1** Sparsified Online Newton (SONew) Algorithm

**Inputs**: $\lambda_t \coloneqq$ coefficient in the update (5),
$\mathcal{G} \coloneqq$ sparsity graph (banded/tridiagonal),
$\epsilon \coloneqq$ damping parameter,
$T \coloneqq$ total number of iterations/mini-batches,
$\eta_t \coloneqq$ step size/learning rate.
**Output**: $w_{T+1}$
1: $H_0 = \epsilon I_d, w_1 = 0$
2: **for** $t \in \{1, \ldots, T\}$ **do**
3:  compute $g_t = \nabla f_t(w_t)$
4:  $H_t \coloneqq H_{t-1} + P_\mathcal{G}(g_t g_t^T / \lambda_t) \in S_n(\mathcal{G})$ with $P_\mathcal{G}$ as in (8).  $\triangleright$ $\mathcal{O}(n)$ time & memory
5:  Get $L, D = $ Sparsified_Inverse $(H_t, \mathcal{G})$, where $X_t = LDL^T$ solves (11).
6:  Compute update $u_t = LDL^T g_t$,
7:  $w_{t+1} = w_t - \eta_t u_t$
8: **end for**
9: **return** $w_{T+1}$

**Algorithm 2** Sparsified_Inverse$(H, \mathcal{G})$ in $\mathcal{O}(n)$ flops

**Inputs**: $H \in S_n(\mathcal{G})$, is as (10).
$\mathcal{G} \coloneqq$ the banded graph of band size $b \ll n$
**Outputs**: lower triangular banded $L \in \mathbb{R}^{n \times n}$ and diagonal matrix $D \in \mathbb{R}^{n \times n}$
1: **function** Sparsified_Inverse$(H, \mathcal{G})$
2:  $L \coloneqq 0, D \coloneqq 0$
3:  $L_{jj} \coloneqq 1, \forall j \in [n]$
4:  **for** $j \in \{1, \ldots, n\}$ **do**  $\triangleright$ parallelizable
5:   Let $H_{jI_j}$ and $H_{I_jI_j}$ be defined as in Section 2, where $I_j = \{j+1, \ldots, j+b\} \cap [n]$,
6:   Solve for $L_{I_jj}$ in the linear system $H_{I_jI_j} L_{I_jj} = -H_{I_jj}$  $\triangleright$ $\mathcal{O}(b^3)$ time.
7:   $D_{jj} \coloneqq 1/(H_{jj} + H_{I_jj}^T L_{I_jj})$
8:  **end for**
9:  **return** $L, D$
10: **end function**

## 3.2 Sparsifying the Preconditioner

To reduce the memory required to maintain and update $X_t$ using (5), we consider the following general problem: find sparse $X \succ 0$ with $\|X\|_0 \le \alpha n$, $\alpha > 1$, such that the objective $D_{\ell d}(X, (X_{t-1}^{-1} + g_t g_t^T / \lambda_t)^{-1})$ is minimized. Due to the $L_0$-norm constraint, this is a non-convex problem, which makes it difficult to solve exactly. Since $L_1$-norm serves as a convex relaxation for the $L_0$ norm, we could use it instead, resulting in the following optimization problem also known as graphical lasso estimator (Friedman et al., 2008):

$$\min_{X \in S_n^{++}} D_{\ell d}\left(X, (X_{t-1}^{-1} + g_t g_t^T / \lambda_t)^{-1}\right) + \lambda \|X\|_1 .$$

The sparsity introduced by $L_1$-norm penalty will reduce the memory usage. However, the time taken to solve the above problem, even with the current best methods (Bollhöfer et al., 2019; Hsieh et al., 2013; Fattahi & Sojoudi, 2019; Zhang et al., 2018), can still be too large (as these methods take several minutes for a matrix of size million), making it impractical to embed in DNN training since preconditioning may need to be done after processing every mini-batch.

In this paper, we take a different direction where we use fixed sparsity pattern constraints, specified by a fixed undirected graph $\mathcal{G}$. To sparsify the solution in (5), we formulate the subproblem

$$X_t = \underset{X \in S_n(\mathcal{G})^{++}}{\arg\min} D_{\ell d}\left(X, (X_{t-1}^{-1} + g_t g_t^T / \lambda_t)^{-1}\right), \tag{6}$$

where $S_n(\mathcal{G})^{++}$ denotes the set of positive definite matrices with the fixed sparsity pattern given by graph $\mathcal{G}$. Note that even for sparsification we use the LogDet measure; thus both the steps (4) and (6) use the same measure.

Algorithm 1 presents one proposed SONew method, which solves (6) using $\mathcal{O}(n)$ time and memory for banded matrices with band size $b$, in particular a tridiagonal matrix, corresponding to a chain graph, is a banded matrix with bandsize 1.

**Maintaining $H_t \in S_n(\mathcal{G})$ in line 4**. Solving the subproblem in (6) naively is impractical since $X_{t-1}^{-1}$ is a dense matrix. However, the structure of the LogDet divergence comes to the rescue; (6) can be expanded as follows:

$$X_t = \underset{X \in S_n(\mathcal{G})^{++}}{\arg\min} -\log\det(X) + \text{Tr}(X(X_{t-1}^{-1} + g_t g_t^T / \lambda_t)). \tag{7}$$

Let us define the projection onto $S_n(\mathcal{G})$, $P_\mathcal{G} : \mathbb{R}^{n \times n} \to \mathbb{R}^{n \times n}$ as:

$$P_\mathcal{G}(M)_{ij} = \begin{cases} M_{ij} & \text{if } (i,j) \in E_\mathcal{G}, \\ 0 & \text{otherwise} \end{cases}, \tag{8}$$

Note that the $\mathrm{Tr}(.)$ term in (7) involves only the non-zero elements of $X \in S_n(\mathcal{G})^{++}$. Hence, (7) can be written as

$$X_t = \underset{X \in S_n(\mathcal{G})^{++}}{\arg\min} -\log\det(X) + \mathrm{Tr}(XP_{\mathcal{G}}(X_{t-1}^{-1} + g_t g_t^T/\lambda_t)), \qquad (9)$$

Computing matrix $X_{t-1}^{-1}$ can be avoided by analyzing optimality condition of (9). Let $g(X)$ be the objective function in (9), then the optimality condition of (9) is $P_{\mathcal{G}}(\nabla g(X)) = 0$. Expanding $g(X)$ gives

$$P_{\mathcal{G}}(X_t^{-1}) = P_{\mathcal{G}}(X_{t-1}^{-1} + g_t g_t^T/\lambda_t) = P_{\mathcal{G}}(X_{t-1}^{-1}) + P_{\mathcal{G}}(g_t g_t^T/\lambda_t),$$
$$H_t = H_{t-1} + P_{\mathcal{G}}(g_t g_t^T/\lambda_t). \qquad (10)$$

Thus we only need to maintain $H_t \in S_n(\mathcal{G})$, which is $H_t = P_{\mathcal{G}}(X_t^{-1})$. This matrix is updated as $H_t = H_{t-1} + P_{\mathcal{G}}(g_t g_t^T/\lambda_t)$, which can be done in $\mathcal{O}(|E_{\mathcal{G}}|)$ memory and time, while computing the matrix $X_t^{-1}$ would have cost $\mathcal{O}(n^2)$. In SONew, this key observation is used to maintain $H_t$ in line line 4.

**Computing $X_t$ in line 5.** Now that $H_t$ is known at every round $t$, we can replace $P_{\mathcal{G}}(X_{t-1}^{-1} + g_t g_t^T/\lambda_t)$ in (9) with $H_t$ as:

$$X_t = \underset{X \in S_n(\mathcal{G})^{++}}{\arg\min} -\log\det(X) + \mathrm{Tr}(XH_t). \qquad (11)$$

For an arbitrary graph $\mathcal{G}$, solving this subproblem might be difficult. Theorems 1 and 2 show *embarrassingly parallelizable* explicit solutions to the subproblem 11 for tridiagonal and banded sparsity patterns. Proofs of Theorems 1 and 2 are given in Appendix A.1.

**Theorem 1** (Explicit solution of (11) for tridiagonal matrix/chain graph). *Let the sparsity structure $\mathcal{G}$ be a chain with edges $E_{\mathcal{G}} = \{(j, j+1) : j \in [n-1]\}$, and let $H \in S_n(\mathcal{G})$, then the solution of (11) is given by $\hat{X} = LDL^T$, where the unit lower triangular matrix $L$ and diagonal matrix $D$ have the following non-zero entries:*

$$L_{jj} = 1, \ L_{j+1j} = -\frac{H_{j+1j}}{H_{j+1j+1}}, \ D_{jj}^{-1} = \left(H_{jj} - \frac{H_{j+1j}^2}{H_{j+1j+1}}\right), j \leq n-1, \ D_{nn}^{-1} = H_{nn} \qquad (12)$$

The time and memory complexity required to compute (12) is $\mathcal{O}(n)$. Note that the computation of (12) can be easily parallelized. This explicit solution can be generalized to banded sparsity structures with band size $b$.

**Theorem 2** (Explicit solution of (11) for banded matrices). *Let the sparsity pattern $\mathcal{G}$ be a banded matrix of band size $b$, i.e. for every vertex $j$, let $I_j = \{j+1, \ldots, j+b\}$, then edges $E_{\mathcal{G}} = (\bigcup_{j=1}^n \{j\} \times I_j) \cap \{(i,j) : i \leq n, j \leq n\}$. Then $X_t = LDL^T$ is the solution of (11) with nonzero entries of $L$ and $D$ defined as follows :*

$$L_{jj} = 1, \ L_{I_j j} = -H_{I_j I_j}^{-1} H_{I_j j}, \ D_{jj}^{-1} = (H_{jj} - H_{I_j j}^T H_{I_j I_j}^{-1} H_{I_j j}), \ 1 \leq j \leq n. \qquad (13)$$

Finding the above solution requires solving $n$ linear systems of size $b$ (which is small) as shown in Algorithm 2, and takes $\mathcal{O}((n-b+1)b^3)$ flops. Since $b \ll n$, the number of flops is $\mathcal{O}(n)$.

### 3.3 Analysis of SONew

The following theorem establishes regret guarantees of SONew in online convex optimization framework mentioned in Section 3.1.

**Theorem 3.** *When $\mathcal{G} = $ tridiagona/chain graph, then Algorithm 1 incurs the following regret under convex losses $f_t$*

$$R_T \leq \mathcal{O}\left(C^{1/2} \cdot T^{3/4} \cdot \left(\frac{1+\beta}{1-\beta}\right)^{1/2} \cdot \left(\sum_{i=1}^n \log\left(1 + \frac{\left\|g_{1:T}^{(i)}\right\|_2^2}{\epsilon}\right) + \sum_{i=1}^{n-1} \log\left(1 - \beta_i^2\right)\right)^{3/4}\right),$$

where $g_{1:T}^{(i)} = [(g_1)_i, \dots (g_T)_i]$ denotes gradient history of $i^{th}$ variable/parameter ,$G_\infty^{(i)} = \|g_{1:T}^{(i)}\|_\infty$, $C = \max_t(\sum_{i=1}^n (w_t - w^*)_i^2 (G_\infty^{(i)})^2)$, $\beta_i = \dfrac{\left\langle g_{1:T}^{(i)}, g_{1:T}^{(i+1)} \right\rangle}{\sqrt{(\epsilon + \left\|g_{1:T}^{(i)}\right\|_2^2) \cdot (\epsilon + \left\|g_{1:T}^{(i+1)}\right\|_2^2)}}$ denotes dot-product between gradient histories of connected parameters in the chain graph, $\beta = \max_{i \in [n-1]} \beta_i$.

SONew is *scale-invariant* to diagonal transformation of gradients; if the gradients sent as feedback to the OCO learner are $\bar{g}_t = \Lambda g_t$, then the iterates for the transformed problem will be $\bar{x}_t = \Lambda^{-1} w_t = \Lambda^{-1} x_{t-1} - \eta(\Lambda^{-1} X_{t-1} \Lambda^{-1})\Lambda(g_{t-1})$, this is due to scale-invariance property of LogDet divergence in Section 2.1,i.e, the transformed preconditioner is $\bar{X}_t = \Lambda^{-1} X_{t-1} \Lambda^{-1}$. Furthermore, the regret bound derived in Theorem 3 for transformed and original problem is approximately same. Our regret bound is data-dependent, since if $\beta_i$ is nearer to 1, then the regret is smaller due to the $\log(1 - \beta_i^2)$ term. This effect is more amplified when $\log(1 + \|g_{1:T}^{(i)}\|_2^2/\epsilon)$ is relatively low compared to $\log(1 - \beta_i^2)$. The proof for Theorem 3 is given in Appendix A.2. We also derive a regret bound with $\mathcal{O}(\sqrt{\kappa}T^{1/2})$ dependency on $T$ in Appendix A.3, where the condition number $\kappa(\text{diag}(H_t)) \leq \kappa$.

## 4 NUMERICAL STABILITY OF SONEW

The matrix $H \in S_n(\mathcal{G})$ given as input to Algorithm 2 should have positive definite submatrices $H_{J_j J_j}$, $J_j = I_j \cup \{j\}$, $j \in [n]$, where $I_j$ is as in Theorem 2, hence, it should have positive schur complements $(H_{jj} - H_{I_j j}^T H_{I_j I_j}^{-1} H_{I_j j})$. But, if the matrices $H_{J_j J_j}$ and $H_{I_j I_j}$ are illconditioned, then the computed value for $(H_{jj} + H_{I_j j}^T L_{I_j j}) = (H_{jj} - H_{I_j j}^T H_{I_j I_j}^{-1} H_{I_j j})$, in line 7 of Algorithm 2, can be zero or negative due to catastrophic cancellation, since finding $L_{I_j j} = -H_{I_j I_j}^{-1} H_{I_j j}$ in floating point arithmetic can result in rounding errors. To understand this issue further, we conduct perturbation analysis in Theorem 4 which establishes a componentwise condition number (Higham, 2002) of the optimization problem in (11) with a tridiagonal sparsity structure $\mathcal{G}$.

**Theorem 4** (Condition number of tridiagonal LogDet subproblem 11). *Let $H \in S_n^{++}$ be such that $H_{ii} = 1$ for $i \in [n]$. Let $\Delta H$ be a symmetric perturbation such that $\Delta H_{ii} = 0$ for $i \in [n]$, and $H + \Delta H \in S_n^{++}$. Let $P_\mathcal{G}(H)$ be the input to 11, where $\mathcal{G}$ is a chain graph, then*

$$\kappa_\infty^{\ell d} \leq \max_{i \in [n-1]} 2/(1 - \beta_i^2) = \hat{\kappa}_\infty^{\ell d}, \tag{14}$$

*where, $\beta_i = H_{ii+1}, \kappa_\infty^{\ell d} :=$ componentwise condition number of (11) for perturbation $\Delta H$.* [1]

So, the tridiagonal LogDet problem with inputs $H$ as mentioned in Theorem 4, has high condition number when $1 - \beta_i^2 = H_{ii} - H_{ii+1}^2/H_{i+1i+1}$ are low and as a result the preconditioner $X_t$ in SONew (Algorithm 1) has high componentwise relative errors. In SONew (1), the $H_t = P_\mathcal{G}(\sum_{s=1}^t g_s g_s^T / \lambda_t)$ generated in line 4 could be such that the matrix $\sum_{s=1}^t g_s g_s^T / \lambda_t$ need not be positive definite and so the schur complements $H_{ii} - H_{ii+1}^2/H_{i+1i+1}$ can be zero, giving an infinite condition number $\kappa_\infty^{\ell d}$ by Theorem 4. The following lemma describes such cases in detail for a more general banded sparsity structure case.

**Lemma 1** (Degenerate inputs to banded LogDet subproblem). *Let $H = P_\mathcal{G}(GG^T)$, where $G \in \mathbb{R}^{n \times T}$ and let $g_{1:T}^{(i)}$ be $i^{th}$ row of $G$, which is gradients of parameter $i$ for $T$ rounds, then $H_{ij} = \left\langle g_{1:T}^{(i)}, g_{1:T}^{(j)} \right\rangle$.*

- *Case 1: For tridiagonal sparsity structure $\mathcal{G}$: if $g_{1:T}^{(j)} = g_{1:T}^{(j+1)}$, then $H_{jj} - H_{jj+1}^2/H_{j+1j+1} = 0$.*

- *Case 2: For $b > 1$ in (13): If $\text{rank}(H_{J_j J_j}) = \text{rank}(H_{I_j I_j}) = b$, then $(H_{jj} - H_{I_j j}^T H_{I_j I_j}^{-1} H_{I_j j}) = 0$ and $D_{jj} = \infty$. If $\text{rank}(H_{I_j I_j}) < b$ then the inverse $H_{I_j I_j}^{-1}$ doesn't exist and $D_{jj}$ is not well-defined.*

If $GG^T = \sum_{i=1}^T g_i g_i$ is a singular matrix, then solution to the LogDet problem might not be well-defined as shown in Lemma 1. For instance, Case 1 can occur when preconditioning the input layer

---

[1]The full version of Theorem 4 along with proof is given in Appendix A.4.

of an image-based DNN with flattened image inputs, where $j^{th}$ and $(j+1)^{th}$ pixel can be highly correlated throughout the dataset. Case 2 can occur in the first $b$ iterations in Algorithm 1 when the rank of submatrices $\text{rank}(H_{I_j I_j}) < b$ and $\epsilon = 0.^2$ We develop Algorithm 3 which is robust to degenerate inputs $H$, given that $H_{ii} > 0$. It finds a subgraph $\tilde{\mathcal{G}}$ of $\mathcal{G}$ for which (13) is well-defined. This is done by removing edges which causes inverse $H_{I_j I_j}^{-1}$ to be singular or $(H_{jj} - H_{I_j j}^T H_{I_j I_j}^{-1} H_{I_j j})$ to be low. Our ablation study in Table 4 demonstrates noticeable improvement in performance Algorithm 3 is used.

**Theorem 5** (Numerically stable algorithm). *Algorithm 3 finds a subgraph $\tilde{\mathcal{G}}$ of $\mathcal{G}$, such that explicit solution for $\tilde{\mathcal{G}}$ in (13) is well-defined. Furthermore, when $\mathcal{G}$ is a tridiagonal/chain graph, the component-wise condition number upperbound in (14) is reduced upon using Algorithm 3, $\hat{\kappa}_{\ell d}^{\tilde{\mathcal{G}}} < \hat{\kappa}_{\ell d}^{\mathcal{G}}$, where $\hat{\kappa}_{\ell d}^{\tilde{\mathcal{G}}}, \hat{\kappa}_{\ell d}^{\mathcal{G}}$ are defined as in Theorem 4 for graphs $\tilde{\mathcal{G}}$ and $\mathcal{G}$ respectively.*

The proofs for Lemma 1 and Theorem 5 are given in Appendix A.4.

---

**Algorithm 3** Numerically stable banded LogDet solution

---

1: **Input:** $\mathcal{G}-$ tridiagonal or banded graph, $H-$ symmetric matrix in $\mathbb{R}^{n \times n}$ with sparsity structure $\mathcal{G}$ and $H_{ii} > 0$, $\gamma-$ tolerance parameter for low schur complements.
2: **Output:** Finds subgraph $\tilde{\mathcal{G}}$ of $\mathcal{G}$ without any degenerate cases from Lemma 1 and finds preconditioner $\hat{X}$ corresponding to the subgraph
3: Let $E_i = \{(i,j) : (i,j) \in E_{\mathcal{G}}\}$ be edges from vertex $i$ to its neighbours in graph $\mathcal{G}$.
4: Let $V_i^+ = \{j : i < j, (i,j) \in E_{\mathcal{G}}\}$ and $V_i^- = \{j : i > j, (i,j) \in E_{\mathcal{G}}\}$, denote positive and negative neighbourhood of vertex $i$.
5: Let $K = \left\{ i : H_{ii} - H_{I_i i}^T H_{I_i I_i}^{-1} H_{I_i i} \text{ is not defined or } \le \gamma, i \in [n] \right\}$
6: Consider a new subgraph $\tilde{\mathcal{G}}$ with edges $E_{\tilde{\mathcal{G}}} = E_{\mathcal{G}} \setminus (\bigcup_{i \in K} E_i \cup (V_i^+ \times V_i^-))$
7: **return** $\hat{X} := \text{SPARSIFIED\_INVERSE}(\tilde{H}_t, \tilde{\mathcal{G}})$, where $\tilde{H}_t = P_{\tilde{\mathcal{G}}}(H_t)$

---

## 5 RELATED WORK

Online Newton method is a second order method in online convex optimization framework with properties such as scale invariance (Luo et al., 2016) and logarithmic regrets in exp-concave and strongly convex functions (Hazan et al., 2007; 2016). However, it has a time complexity of $\mathcal{O}(n^2)$, making it infeasible for large $n$. A diagonal version of this method SC-Adagrad (Mukkamala & Hein, 2017) was proposed to make the online Newton method scalable for deep-learning. SC-Adagrad is equivalent to setting the sparsity pattern $\mathcal{G}$ in equation 11 as a null graph, furthermore, setting $\mathcal{G}$ as a complete graph will result in online Newton method. However, introduction of LogDet divergence measure in SONew allows us to set different sparsity graphs as $\mathcal{G}$ such as banded graph with band-size $b$, for which our preconditioning process is more computationally efficient with a time complexity of $\mathcal{O}(b^3(n-b+1))$ compared to online-newton method $\mathcal{O}(n^2)$. As discussed in Theorem 3, SONew also utilizes correlation among gradients of parameters to improve convergence, in contrast to diagonal preconditioners such as SC-Adagrad.

Shampoo (Gupta et al., 2018; Anil et al., 2020) uses Kronecker factored preconditioners to reduce the memory and time complexity from $\mathcal{O}(n^2)$ to $\mathcal{O}(d_1^2 + d_2^2)$ and $\mathcal{O}(d_1^3 + d_2^3)$ respectively, where $n = d_1 d_2$ denotes number of parameters for a linear layer of dimensions $d_1 \times d_2$. The time complexity of matrix inversion takes a heavy toll in Shampoo's compute time even with the Kronecker product assumption on the preconditioner, whereas, our method has a time complexity of $\mathcal{O}(b^3 d_1 d_2)$ quadratic in dimensions of the linear layer (note that $b = 1$ for tridiagonal structure). Furthermore, precision lost in matrix inversion can be large due to high condition number matrices occurring in training (Anil et al., 2022; 2020).

LogDet problem in equation 11 is closely related to the Maximum Determinant Matrix Completion (MDMC) (Andersen et al., 2013; Vandenberghe et al., 2015). The MDMC problem is the dual of LogDet problem (11), and has explicit solutions for chordal graphs (Andersen et al., 2013). Thus the explicit solutions in (13) are the same as the ones proved in Andersen et al. (2013). Also, we noticed that the tridiagonal explicit solution has been used previously in KFAC (Martens & Grosse, 2015) in the context of a gaussian graphical model interpretation of gradients. In this paper we also analyze conditioning and degenerate cases of these explicit solutions (which can occur frequently in

---

²This is proved in Appendix A.4

DNN training) and develop algorithms to avoid such cases. In addition, we have regret guarantees connecting these explicit solutions to regret in online convex optimization problem setup. There is prior work (Luo et al., 2016; 2019) in reducing the complexity - $\mathcal{O}(n^2)$ flops of Online Newton Step (ONS) to $\mathcal{O}(n)$ flops using sketching. These ONS variants maintain a low rank approximation of $H_t$ (as in Algorithm 1) and updating it with a new gradient $g_t$ at every iteration requires conducting SVD (Luo et al., 2019)/orthonormalization (Luo et al., 2016) of a tall and thin matrix in $\mathbb{R}^{n \times r}$, where $r$ denotes the rank of approximation of $H_t$. Our proposed method (Algorithm 1) has a more parallelizable update $H_t := H_{t-1} + P_\mathcal{G}(g_t g_t^T)$ making it more suitable for DNN training.

## 6 EXPERIMENTAL RESULTS

In this section we describe our experiments on Autoencoder benchmark (Schmidhuber, 2015) using MNIST dataset (Deng, 2012). Our results on larger benchmarks are given in Appendix A.5. We compare SONew against commonly used first order methods including SGD (Kiefer & Wolfowitz, 1952)), SGD with Momentum (Qian, 1999), Nesterov (Nesterov, 1983), Adagrad (Duchi et al., 2011a), Adam (Kingma & Ba, 2014), and Rmsprop (Tieleman & Hinton, 2012). We also compare with Shampoo (Gupta et al., 2018), a state of the art second-order optimizer used in practice. Computing preconditioner at every step in shampoo could be infeasible, instead it is computed every $t$ steps - referred to as Shampoo($t$) in the experiments.

For SONew, we use exponentially moving average (EMA) for both first order and second order statistics. EMA is commonly used in adaptive optimizers for deep-learning training Kingma & Ba (2014); Qian (1999). Let $\beta_1, \beta_2 \in [0, 1]$ be the coefficients for EMA and $g_t \in \mathbb{R}^n$ be the gradient at $t^{th}$ iteration/mini-batch. Let $\mu_t \in \mathbb{R}^n$ be the first order gradient statistic, and $H_t \in S_n(\mathcal{G})$ be the second order gradient statistic as in (10). Then, the following modification to update rules are made to in the SONew implementation.

$$\mu_t = \beta_1 \mu_{t-1} + (1 - \beta_1)g_t, \quad H_t = \beta_2 H_{t-1} + (1 - \beta_2)g_t g_t^T.$$

Furthermore, $\mu_t$ is used in $w_{t+1} = w_t - \eta_t X_t \mu_t$, replacing the gradient $g_t$, similar to Kingma & Ba (2014). As discussed in Section 3.2 we need to store the values of $H_t$ only corresponding to the sparsity pattern, hence SONew uses $\mathcal{O}(n)$ space. The updates above get computed in parallel in $\mathcal{O}(1)$ time. Moreover, we use Algorithm 3 to make SONew numerically stable. We also use grafting (Agarwal et al., 2022), a technique used to transfer step size between optimization algorithms. Specifically, given an update $v_1$ of Optimizer-1 and $v_2$ of Optimizer-2, grafting allows to use the direction suggested by Optimizer-2 with step size suggested by Optimizer-1. The final update is given by $\frac{\|v_1\|}{\|v_2\|} \cdot v_2$. Grafting has been shown to take advantage of a tuned optimizer step size and improve performance. In our case, we use Adam grafting - using Adam optimizer step size $\|v_1\|$ with SONew direction $v_2/\|v_2\|$.

We use three sparsity patterns for SONew - a) diagonal sparsity, resulting in a diagonal preconditioner similar to adaptive first order methods like Adam and Adagrad; b) tridiagonal sparsity, corresponding to a chain graph; and c) banded sparsity, represented by "band-$k$" in tables and figures for band size of $k$.

All the baselines and SONew are trained for 100 epochs and use the train set of MNIST dataset containing $60k$ points. We use a standard sized (2.72M parameters) Autoencoder with layer sizes: [1000, 500, 250, 30, 250, 500, 1000] and tanh non-linearity. Batch size is fixed at 1000 and we use learning rate schedule with a linear warmup of 5 epochs followed by linear decay towards 0. Over $2k$ hyperparameters are searched over for each experiment using a Bayesian optimization package. The search space for each optimizer is mentioned in Appendix A.5.

From the float32 experiments in Table 1 we observe that among first order methods, diag-SONew performs the best while taking same amount of time. Increasing the number of edges in the sparsity graph to tridiag or banded with band size 4 enhances the performance further. Tridiag-SONew performs $4\times$ faster than Shampoo at a marginal cost to the loss - even when Shampoo updates preconditioner once every 20 steps. When dealing with large scale models like Resnet50 (He et al., 2015a), Shampoo requires updating preconditioner much more often (Anil et al., 2019), whereas our method is also promising in such scenarios. We leave comparison of SONew with Shampoo on such large benchmarks as a future work. To show efficacy of SONew at lower precision, we conduct bfloat16 experiments. We notice in Table 2 that diag-SONew performs the best in first order methods, just like in float32. Moreover SONew undergoes the least degradation in performance

Table 1: **float32 experiments on Autoencoder benchmark.** "tridiag" repersents tridiag-SONew, and "band-4" represents banded-SONew with band size 4. We observe that diag-SONew performs the best among all first order methods while taking similar time. tridiag and band-4 perform significantly better than first order methods while requiring similar linear space and being marginally slower. Shampoo performs best but takes $\mathcal{O}(d^3)$ time for computing preconditioner of a linear layer of size $d \times d$, whereas our methods take $\mathcal{O}(d^2)$ to find the precondioner.

| Optimizer | First Order Methods | | | | | | | Second Order Methods | | |
|---|---|---|---|---|---|---|---|---|---|---|
| | SGD | Nesterov | Adagrad | Momentum | RMSProp | Adam | diag-SONew | Shampoo(20) | tridiag | band-4 |
| Train CE loss | 67.654 | 59.087 | 54.393 | 58.651 | 53.330 | 53.591 | 53.025 | 50.702 | 51.723 | 51.357 |
| Time(s) | 62 | 102 | 62 | 67 | 62 | 62 | 63 | 371 | 87 | 251 |

Table 2: **bfloat16 experiments on Autoencoder benchmark.** diag-SONew performs the best among all first order methods, while degrading only marginally (0.26 absolute difference) compared to float32 performance. tridiag-SONew and banded-SONew holds similar observations as well. Shampoo performs the best but has a considerable drop (0.70) in performance compared to float32 due to using matrix inverse, and is slower due to its cubic time complexity for computing preconditioners. Shampoo implementation uses 16-bit quantization to make it work in 16-bit setting, leading to further slowdown. Hence the running time in bfloat16 is even higher than in float32.

| Optimizer | First Order Methods | | | | | | | Second Order Methods | | |
|---|---|---|---|---|---|---|---|---|---|---|
| | SGD | Nesterov | Adagrad | Momentum | RMSProp | Adam | diag-SONew | Shampoo(20) | tridiag | band-4 |
| Train CE loss | 80.454 | 72.975 | 68.854 | 70.053 | 53.743 | 54.328 | 53.29 | 51.401 | 51.937 | 51.84 |
| Train time(s) | 36 | 43 | 37 | 36 | 37 | 38 | 44 | 1245 | 70 | 502 |

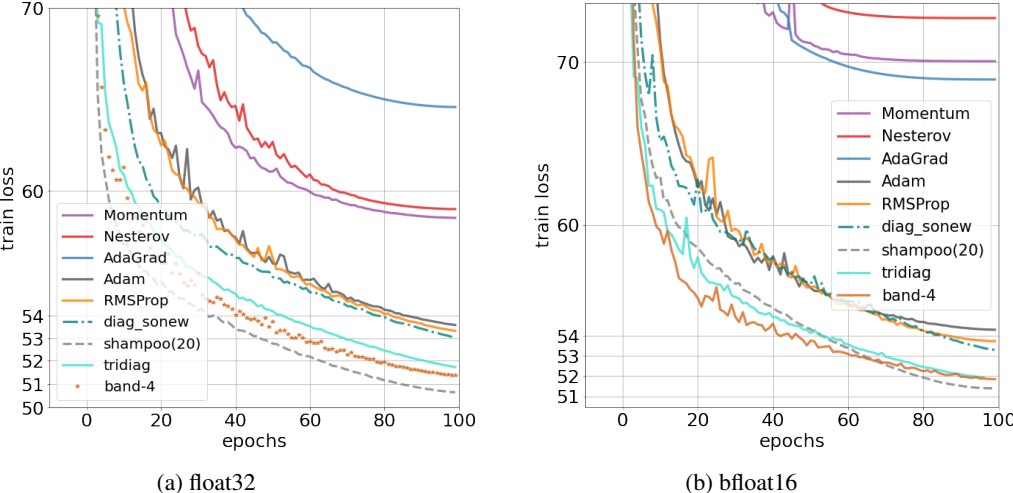

(a) float32          (b) bfloat16

Figure 1: (a) Comparison of SONew (tridiag, band-4) with first-order optimizers and Shampoo (second-order). Left (a) uses float32 training and right (b) uses bfloat16 training. We observe that tridiag and banded SONew have better convergence compared to shampoo in lower precision in early stages, and performs better than all first order methods in float32 and bfloat16.

compared to all other optimizers. We also provide an ablation study on effect of using Algorithm 3 on training loss in the appendix.

## 7    CONCLUSIONS AND FUTURE WORK

In this paper we have introduced a computationally efficient sparse preconditioner. Our algorithm arises from a novel regret bound analysis using LogDet Divergence, and furthermore we make it numerically stable. Experimental results on the Autoencoder benchmark confirm the effectiveness of SONew in both float32 as well as bfloat16 precision. In the future, one can explore different sparsity graphs for which efficient solutions exist for the LogDet subproblem (11).

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

## A APPENDIX

### A.1 PROPERTIES OF LOGDET SUBPROBLEM

*Proof of Theorem 2*

The optimality condition of (11) is

$$P_\mathcal{G}(X^{-1}) = P_\mathcal{G}(H)$$

Let $Z = L^{-T}D^{-1}L^{-1}$, then $P_\mathcal{G}(Z) = H$

$$ZL = L^{-T}D^{-1} \implies ZLe_j = L^{-T}D^{-1}e_j$$

Let $J_j = I_j \cup j$, then select $J_j$ indices of vectors on both sides of the second equality.

$$\begin{bmatrix} Z_{jj} & Z_{jI_j} \\ Z_{I_jj} & Z_{J_jJ_j} \end{bmatrix} \begin{bmatrix} 1 \\ L_{I_j} \end{bmatrix} = \begin{bmatrix} 1/d_{jj} \\ 0 \end{bmatrix} \tag{15}$$

Note that $L^{-T}$ is an upper triangular matrix with ones in the diagonal hence $J_j^{th}$ block of $L^{-T}e_j$ will be $[1, 0, 0, \ldots]$. Also, since $P_{\mathcal{G}}(Z) = H$

$$\begin{bmatrix} Z_{jj} & Z_{jI_j} \\ Z_{I_jj} & Z_{J_jJ_j} \end{bmatrix} = \begin{bmatrix} H_{jj} & H_{jI_j} \\ H_{I_jj} & H_{J_jJ_j} \end{bmatrix}$$

Substituting this in the linear equation 15

$$\begin{bmatrix} H_{jj} & H_{jI_j} \\ H_{I_jj} & H_{J_jJ_j} \end{bmatrix} \begin{bmatrix} 1 \\ L_{I_j} \end{bmatrix} = \begin{bmatrix} 1/d_{jj} \\ 0 \end{bmatrix}$$

$$\begin{bmatrix} H_{jj} & H_{jI_j} \\ H_{I_jj} & H_{J_jJ_j} \end{bmatrix} \begin{bmatrix} d_{jj} \\ d_{jj} \cdot L_{I_j} \end{bmatrix} = \begin{bmatrix} 1 \\ 0 \end{bmatrix}$$

$$H_{jj}d_{jj} + d_{jj}H_{I_jj}^T L_{I_jj} = 1$$

$$H_{I_jj}d_{jj} + d_{jj}H_{I_jI_j}L_{I_jj} = 0$$

The lemma follows from solving the above equations. Note that here we used that lower triangular halves of matrices $L$ and $H$ have the same sparsity patterns, which follows from the fact that banded graph is chordal and has a perfect elimination order $[1, 2, \ldots, n]$.

*Proof of Theorem 1* The proof follows trivially from Theorem 1, when $b$ is set to 1.

### A.2 REGRET BOUND ANALYSIS

*Proof of Theorem 3.*

To upperbound the regret we prove the following lemma about regret decompsition

**Lemma 2** ( Hazan et al. (2016) ). *In the OCO problem setup, if a prediction $w_t \in \mathbb{R}^n$ is made at round $t$ and is updated as $w_{t+1} := w_t - \eta X_t g_t$ using a preconditioner matrix $X_t \in S_n^{++}$*

$$R_T \leq \frac{1}{2\eta} \cdot (\|w_1 - w^*\|_{X_1^{-1}}^2 - \|w_{T+1} - w^*\|_{X_T^{-1}}) \tag{16}$$

$$+ \frac{1}{2\eta} \cdot \sum_{t=1}^{T-1} (w_{t+1} - w^*)^T (X_{t+1}^{-1} - X_t^{-1})(w_{t+1} - w^*) \tag{17}$$

$$+ \sum_{t=1}^{T} \frac{\eta}{2} \cdot g_t^T X_t g_t \tag{18}$$

*Proof.*

$$\|w_{t+1} - w^*\|_{X_t^{-1}}^2 = \|w_t - \eta X_t g_t - w^*\|_{X_t^{-1}}^2$$

$$= \|w_t - w^*\|_{X_t^{-1}}^2 + \eta^2 g_t^T X_t g_t - 2\eta(w_t - w^*)^T g_t$$

$$\implies 2\eta(w_t - w^*)^T g_t = \|w_t - w^*\|_{X_t^{-1}}^2 - \|w_{t+1} - w^*\|_{X_t^{-1}}^2 + \eta^2 g_t^T X_t g_t$$

$$\square$$

Using the convexity of $f_t$, $f_t(w_t) - f_t(w^*) \leq (w_t - w^*)^T g_t$, where $g_t = \Delta f_t(w_t)$ and summing over $t \in [T]$

$$R_T \leq \sum_{t=1}^{T} \frac{1}{2\eta} \cdot \left( \|w_t - w^*\|_{X_t^{-1}}^2 - \|w_{t+1} - w^*\|_{X_t^{-1}}^2 \right) + \frac{\eta}{2} \cdot g_t^T X_t g_t \tag{19}$$

The first summation can be decomposed as follows

$$\sum_{t=1}^{T} \left( \|w_t - w^*\|_{X_t^{-1}}^2 - \|w_{t+1} - w^*\|_{X_t^{-1}}^2 \right) = \left( \|w_1 - w^*\|_{X_1^{-1}}^2 - \|w_{T+1} - w^*\|_{X_T^{-1}}^2 \right)$$

$$+ \sum_{t=1}^{T-1} (w_{t+1} - w^*)^T (X_{t+1}^{-1} - X_t^{-1})(w_{t+1} - w^*)$$

Substituting the above identity in the Equation (19) proves the lemma.

Let $R_T \leq T_1 + T_2 + T_3$, where

- $T_1 = \frac{1}{2\eta} \cdot (\|w_1 - w^*\|^2_{X_1^{-1}} - \|w_{T+1} - w^*\|_{X_T^{-1}})$
- $T_2 = \frac{1}{2\eta} \cdot \sum_{t=1}^{T-1} (w_{t+1} - w^*)^T (X_{t+1}^{-1} - X_t^{-1})(w_{t+1} - w^*)$
- $T_3 = \sum_{t=1}^{T} \frac{\eta}{2} \cdot g_t^T X_t g_t$

The following lemmas will be used to bound $T_1$, $T_2$, $T_3$.

**Lemma 3.** *If $\mathcal{G}$ = chain/tridiagonal graph and $\hat{X} = \arg\min_{X \in S_n(\mathcal{G})^{++}} \mathrm{D}_{\ell d}(X, H^{-1})$, then the inverse $\hat{X}^{-1}$ takes the following expression*

$$(\hat{X}^{-1})_{ij} = \begin{cases} H_{ij} & |i - j| \leq 1 \\ \frac{H_{ii+1} H_{i+1i+2} \dots H_{j-1j}}{H_{i+1i+1} \dots H_{j-1j-1}} \end{cases} \tag{20}$$

*Proof.*

$$\hat{X}^{-1} \hat{X}^{(j)} = e_j$$

Where $\hat{X}^{(j)}$ is the $j^{th}$ column of $\hat{X}$. Let $\hat{Y}$ denote the right hand side of Equation (20).

$$\begin{aligned} (\hat{Y}\hat{X})_{jj} &= \hat{X}_{jj}\hat{Y}_{jj} + \hat{X}_{j-1j}\hat{Y}_{j-1j} + \hat{X}_{jj+1}\hat{Y}_{jj+1} \\ &= \hat{X}_{jj}H_{jj} + \hat{X}_{j-1j}H_{j-1j} + \hat{X}_{jj+1}H_{jj+1} \\ &= 1 \end{aligned}$$

The third equality is by using the following alternative form of Equation (12):

$$(\hat{X}^{(1)})_{i,j} = \begin{cases} 0 & \text{if } j - i > 1 \\ \frac{-H_{i,i+1}}{(H_{ii}H_{i+1,i+1} - H_{i+1,i+1}^2)} & \text{if } j = i + 1 \\ \frac{1}{H_{ii}} \left(1 + \sum_{j \in \mathrm{neig}_{\mathcal{G}}(i)} \frac{H_{ij}^2}{H_{ii}H_{jj} - H_{ij}^2}\right) & \text{if } i = j \end{cases}, \tag{21}$$

where $i < j$. Similarly, the offdiagonals of $\hat{Y}\hat{X}$ can be evaluated to be zero as follows.

$$\begin{aligned} (\hat{Y}\hat{X})_{ij} &= \hat{Y}_{ij}\hat{X}_{jj} + \hat{Y}_{ij-1}\hat{X}_{j-1j} + \hat{Y}_{ij+1}\hat{X}_{j+1j} \\ &= \hat{Y}_{ij}\hat{X}_{jj} + \hat{Y}_{ij}\frac{H_{j-1j-1}}{H_{j-1j}} + \hat{Y}_{ij}\frac{H_{jj+1}}{H_{jj}}\hat{X}_{j+1j} \\ &= 0 \end{aligned}$$

$\square$

**Lemma 4.** *Let $y \in \mathbb{R}^n$, $\beta = \max_t \max_{i \in [n-1]}(H_t)_{ii+1}/\sqrt{(H_t)_{ii}(H_t)_{i+1i+1}} < 1$, $C = \sum_{i=1}^{n} y_i^2 (G_\infty^{(i)})^2$, where $g_{1:T}^{(i)} = [(g_1)_i, \dots (g_T)_i]$ and $G_\infty^{(i)} = \|g_{1:T}^{(i)}\|_\infty$, then*

$$y^T X_t^{-1} y \leq (Ct + \epsilon \|y\|_2^2) \left(\frac{1 + \beta}{1 - \beta}\right)$$

*Proof.* Let $\tilde{X}_t^{-1} = \mathrm{diag}(H_t)^{-1/2} \hat{X}_t \mathrm{diag}(H_t)^{-1/2}$

$$y^T X_t^{-1} y \leq \left\|\mathrm{diag}(H_t)^{1/2} y\right\|_2^2 \left\|\tilde{X}_t^{-1}\right\|_2 \tag{22}$$

Using the identity of spectral radius $\rho(X) \leq \|X\|_\infty$ and since $\tilde{X}$ is positive definite, $\left\|\tilde{X}_t^{-1}\right\|_2 \leq \|\tilde{X}_t^{-1}\|_\infty$

$$\begin{aligned} \left\|\tilde{X}_t^{-1}\right\|_2 &\leq \max_i \left\{\sum_j \left|(\tilde{X}_t^{-1})_{ij}\right|\right\} \\ &\leq 1 + 2(\beta + \beta^2 + \dots) \\ &\leq \frac{1 + \beta}{1 - \beta} \end{aligned}$$

The second inequality is using Lemma 3. Using $(H_t)_{ii} = \left\| g_{1:T}^{(i)} \right\|_2^2 + \epsilon$ in Equation (22) will give the lemma. $\qquad\square$

**Lemma 5** (Upperbound of $T_1$). $T_1 \leq \frac{(C + \epsilon D_2^2)}{2\eta} \cdot \frac{1+\beta}{1-\beta}$, where $D_2 = \max_{t \in [T]} \|w_t - w^*\|_2$, where $C = \max_t (\sum_{i=1}^n (y_i^{(t)})^2 (G_\infty^{(i)})^2)$.

*Proof.* Since $X_T$ is positive definite

$$
\begin{aligned}
T_1 &\leq \frac{\|w_1 - w^*\|_{X_1^{-1}}^2}{2\eta} \\
&\leq \frac{(y^{(1)})^T X_1^{-1} y^{(1)}}{2\eta}
\end{aligned}
$$

Using Lemma 4 proves the lemma. $\qquad\square$

**Upperbounding $T_2$**

Let $y_t = w_t - w^*$, $P_t = y_t^T X_t^{-1} y_t$, and $Q_t = y_t^T X_{t-1}^{-1} y_t$. Let $P = [P_2, \cdots, P_T]$ and $Q = [Q_2, \cdots, Q_T]$ be the two array representations. Then,

$$
\begin{aligned}
2\eta T_2 &= \sum_{i=1}^{T-1} y_{t+1}^T \left( X_{t+1}^{-1} - X_t^{-1} \right) y_{t+1} \\
&= \sum_{i=2}^{T} y_t^T \left( X_t^{-1} - X_{t-1}^{-1} \right) y_t \\
&= \sum_{i=2}^{T} y_t^T X_t^{-1} y_t - y_t^T X_{t-1}^{-1} y_t \\
&= \sum_{i=2}^{T} P_t - Q_t \\
&= \|P - Q\|_1 \qquad (23)
\end{aligned}
$$

In order to upper bound this, we derive here a generalized version of Pinsker's inequality for our setting.

**Lemma 6.** *Let $A = [A_1, \cdots, A_T]$ and $B = [B_1, \cdots, B_T]$ be two T-length arrays. Let $B_i, A_i > 0 \; \forall i \in [1, T+1]$. Then,*

$$
\|A - B\|_1^2 \leq \frac{2}{3} \left( \sum_{i=1}^{T} (2A_i + B_i) \right) D_{KL}(B, A)
$$

*where the generalized KL-Divergence $D_{KL}(B, A) = \sum_{i=1}^{T} \left( B_i \log \left( \frac{B_i}{A_i} \right) - B_i + A_i \right)$.*

*Proof.* This proof is a mock up of Pinsker's inequality for probability spaces. First step involves using the following identity, for $t \geq -1$

$$
(1+t) \log(1+t) - t \geq \frac{1}{2} \cdot \frac{t^2}{1 + \frac{t}{3}}
$$

Let $t_i = \frac{B_i}{A_i} - 1$. Then, $-1 \leq t_i$. Therefore,

$$D_{KL}(B, A) = \sum_{i=1}^{T} \left( B_i \log\left(\frac{B_i}{A_i}\right) - B_i + A_i \right)$$

$$= \sum_{i=1}^{T} A_i \left( \frac{B_i}{A_i} \log\left(\frac{B_i}{A_i}\right) - \frac{B_i}{A_i} + 1 \right)$$

$$= \sum_{i=1}^{T} A_i \left( (t_i + 1) \log(t_i + 1) - t_i \right)$$

$$\geq \sum_{i=1}^{T} \frac{A_i t_i^2}{2\left(1 + \frac{t_i}{3}\right)}$$

$$D_{KL}(B, A) = \sum_{i=1}^{T} \frac{A_i t_i^2}{2\left(1 + \frac{t_i}{3}\right)} \cdot \frac{\sum_{j=1}^{T} A_j (1 + \frac{t_j}{3})}{\sum_{k=1}^{T} A_k (1 + \frac{t_k}{3})}$$

$$\geq \frac{\left( \sum_{i=1}^{T} A_i |t_i| \right)^2}{2 \sum_{k=1}^{T} A_k (1 + \frac{t_k}{3})} \qquad \text{(Using Cauchy-Schwartz)}$$

$$= \frac{\left( \sum_{i=1}^{T} |A_i - B_i| \right)^2}{\frac{2}{3} \sum_{k=1}^{T} (2A_k + B_k)}$$

$$= \frac{\|A - B\|_1^2}{\frac{2}{3} \sum_{j=1}^{T} (2A_i + B_i)}$$

$\square$

Using the above to bound Equation (23) with $B := P$ and $A := Q$, we get

$$2\eta T_2 = \|P - Q\|_1 \leq \sqrt{\frac{2}{3} \left( \sum_{t=2}^{T} (2Q_t + P_t) \right) D_{KL}(P, Q)}$$

$$= \sqrt{\frac{2}{3} \left( \sum_{t=2}^{T} (2Q_t + P_t) \right) \sum_{t=2}^{T} D_{KL}(P_i, Q_i)}$$

$$= \sqrt{\frac{2}{3} \left( \sum_{t=2}^{T} (2Q_t + P_t) \right) \sum_{t=2}^{T} D_{KL}(y_t^T X_t^{-1} y_t, y_t^T X_{t-1}^{-1} y_t)}$$

$$= \sqrt{\frac{2}{3} \left( \sum_{t=2}^{T} (2Q_t + P_t) \right) \sum_{t=2}^{T} D_{KL}(\bar{y}_t^T \bar{X}_t^{-1} \bar{y}_t, \bar{y}_t^T \bar{X}_{t-1}^{-1} \bar{y}_t)}$$

$$\leq \sqrt{\frac{2}{3} \left( \sum_{t=2}^{T} (2Q_t + P_t) \right) \cdot \sum_{t=2}^{T} \|\bar{y}_t\|_2^2 \cdot D_{vN}(\bar{X}_t^{-1}, \bar{X}_{t-1}^{-1})} \qquad (24)$$

Here, $\bar{y}_t = \Lambda_t y_t$, $\bar{X}_t = \Lambda_t X_t \Lambda_t$, $\bar{X}_{t-1} = \Lambda_t X_{t-1} \Lambda_t$ $D_{vN}$ is von-Neuman Divergence, and last inequality is from Lindblad (1975). We upper bound Equation (24) using the following lemma.

**Lemma 7.** *Let $A, B$ be two PD matrices. Then,*

$$D_{vN}(A, B) \leq \lambda_{max}(A) \cdot D_{\ell d}(B, A)$$

*Proof.* Let $A = V\Lambda V^T$ and $B = U\Theta U^T$

$$D_{vN}(A,B) = \sum_{i,j}(v_i^T u_j)\left(\lambda_i \log\left(\frac{\lambda_i}{\theta_j}\right) - \lambda_i + \theta_j\right)$$

$$= \sum_{i,j}(v_i^T u_j)\lambda_i\left(\log\left(\frac{\lambda_i}{\theta_j}\right) - 1 + \frac{\theta_j}{\lambda_i}\right)$$

$$\leq \max_i \lambda_i \sum_{i,j}(v_i^T u_j)\left(-\log\left(\frac{\theta_j}{\lambda_i}\right) - 1 + \frac{\theta_j}{\lambda_i}\right)$$

$$= \lambda_{max}(A) \cdot \mathrm{D}_{\ell d}(B,A)$$

$\square$

Combining the above result, we get

$$T_2 = \frac{1}{2\eta} \cdot \|P - Q\|_1$$

$$\leq \frac{1}{2\eta} \cdot \sqrt{\frac{2}{3}\left(\sum_{t=2}^{T}(2Q_t + P_t)\right) \cdot \sum_{t=2}^{T}\|\bar{y}_t\|_2^2 \left\|\bar{X}_t^{-1}\right\|_2 \cdot \mathrm{D}_{\ell d}(\bar{X}_{t-1}^{-1}, \bar{X}_t^{-1})}$$

$$\leq \frac{1}{2\eta} \cdot \sqrt{\frac{2}{3}\left(\sum_{t=2}^{T}(2Q_t + P_t)\right) \cdot \sum_{t=2}^{T}\|\bar{y}_t\|_2^2 \left\|\bar{X}_t^{-1}\right\|_2 \cdot \mathrm{D}_{\ell d}(\bar{X}_{t-1}^{-1}, \bar{X}_t^{-1})}$$

The second inequality is due to scale invariance of LogDet divergence. If we set $\Lambda_t = \mathrm{diag}(H_t)^{1/2}$, then using Lemma 4

$$T_2 \leq \frac{1}{2\eta} \cdot \sqrt{\frac{2}{3}\left(\sum_{t=2}^{T}(2Q_t + P_t)\right) \cdot \sum_{t=2}^{T}(Ct + \epsilon\|y_t\|_2^2)\left(\frac{1+\beta}{1-\beta}\right) \cdot \mathrm{D}_{\ell d}(X_{t-1}^{-1}, X_t^{-1})}$$

$$\leq \frac{1}{2\eta} \cdot \sqrt{\frac{2}{3}\left(\sum_{t=2}^{T}(2Q_t + P_t)\right) \cdot (CT + \epsilon D_2^2)\left(\frac{1+\beta}{1-\beta}\right) \cdot \sum_{t=2}^{T}\mathrm{D}_{\ell d}(X_{t-1}^{-1}, X_t^{-1})}$$

$$\leq \frac{1}{2\eta} \cdot (CT + \epsilon D_2^2)\left(\frac{1+\beta}{1-\beta}\right) \cdot \sqrt{\frac{2}{3} \cdot T \cdot \sum_{t=2}^{T}\mathrm{D}_{\ell d}(X_{t-1}^{-1}, X_t^{-1})} \tag{25}$$

where, $D_2 = \max_t \|y_t\|_2$, the second and third inequality is using Lemma 4. Now to bound the $\mathrm{D}_{\ell d}(X_{t-1}^{-1}, X_t^{-1})$ term in the above equation, we develop the following lemma.

**Lemma 8.** $\mathrm{Tr}(X_t H_t) = n$, $\forall t \in \{2, \ldots, T\}$ *in Algorithm 1 with any* $\mathcal{G}$

*Proof.* $P_{\mathcal{G}}(X_t^{-1}) = H_t$ is an optimality condition of LogDet subproblem Equation (11)

$$\mathrm{Tr}(X_t H_t) = \mathrm{Tr}(X_t P_{\mathcal{G}}(X_t^{-1}))$$

$$= \mathrm{Tr}(X_t X_t^{-1})$$

$$= n$$

The second equality is because $X_t$ follows the sparsity graph $\mathcal{G}$. $\square$

**Lemma 9.**

$$\sum_{t=2}^{T}\mathrm{D}_{\ell d}(X_{t-1}^{-1}, X_t^{-1}) \leq \log\left(\frac{\det\left(X_T^{-1}\right)}{\det\left(X_1^{-1}\right)}\right)$$

$$\leq \sum_{i=1}^{n}\log\left(1 + \frac{\left\|g_{1:T}^{(i)}\right\|_2^2}{\epsilon}\right) + \sum_{i=1}^{n-1}\log\left(1 - \beta_i^2\right)$$

where $\beta_i = \dfrac{\left\langle g_{1:T}^{(i)}, g_{1:T}^{(i+1)} \right\rangle}{\sqrt{\left(\epsilon + \left\| g_{1:T}^{(i)} \right\|_2^2\right) \cdot \left(\epsilon + \left\| g_{1:T}^{(i+1)} \right\|_2^2\right)}}$

*Proof.* LogDet divergence is defined as follows

$$\mathrm{D}_{\ell d}(X_{t-1}^{-1}, X_t^{-1}) = -\log\left(\frac{\det\left(X_{t-1}^{-1}\right)}{\det\left(X_t^{-1}\right)}\right) + \mathrm{Tr}(X_{t-1}^{-1} X_t) - n$$

The second two terms on right hand side can be simplified analyzed as follows.

$$
\begin{aligned}
\mathrm{Tr}(X_{t-1}^{-1} X_t) - n &= \mathrm{Tr}(X_{t-1}^{-1} X_t - X_t^{-1} X_t) \\
&= \mathrm{Tr}((X_{t-1}^{-1} - X_t^{-1}) X_t) \\
&= -g_t^T X_t g_t \\
&< 0
\end{aligned}
\tag{26}
$$

Thus

$$
\begin{aligned}
\sum_{t=2}^{T} \mathrm{D}_{\ell d}\left(X_{t-1}^{-1}, X_t^{-1}\right) &\leq \sum_{t=2}^{T} -\log\left(\frac{\det\left(X_{t-1}^{-1}\right)}{\det\left(X_t^{-1}\right)}\right) \\
&\leq \log\left(\frac{\det\left(X_T^{-1}\right)}{\det\left(X_1^{-1}\right)}\right)
\end{aligned}
$$

Since LogDet divergence is positive

$$
\begin{aligned}
\mathrm{D}_{\ell d}(X_{t-1}^{-1}, X_t^{-1}) &= -\log\left(\frac{\det\left(X_{t-1}^{-1}\right)}{\det\left(X_t^{-1}\right)}\right) + \mathrm{Tr}(X_{t-1}^{-1} X_t) - n \\
&\geq 0 \\
\implies \log\left(\frac{\det\left(X_t^{-1}\right)}{\det\left(X_{t-1}^{-1}\right)}\right) &\geq 0
\end{aligned}
\tag{27}
$$

The second inequality uses Equation (26). Now, using Equation (13), which is in cholesky decomposition format,

$$
\begin{aligned}
\log(\det\left(X_T^{-1}\right)) &= \sum_{i=1}^{n-1} \log\left((H_T)_{ii} - (H_T)_{ii+1}^2/(H_T)_{i+1i+1}\right) + \log((H_T)_{nn}) \\
&\leq \sum_{i=1}^{n-1} \log(1 - \beta_i^2) + \sum_{i=1}^{n} \log((H_T)_{ii})
\end{aligned}
$$

Using the above, we can expand the log deteriminatn difference.

$$
\begin{aligned}
\log\left(\frac{\det\left(X_T^{-1}\right)}{\det\left(X_1^{-1}\right)}\right) &\leq \log\left(\frac{\det\left(X_T^{-1}\right)}{\det\left(X_0^{-1}\right)}\right) \\
&\leq \sum_{i=1}^{n} \log\left(1 + \frac{\left\| g_{1:T}^{(i)} \right\|_2^2}{\epsilon}\right) + \sum_{i=1}^{n-1} \log\left(1 - \beta_i^2\right)
\end{aligned}
\tag{28}
$$

Let $X_0 = \arg\min_{X \in S_n(\mathcal{G})^{++}} \mathrm{D}_{\ell d}(X, H_0^{-1})$, the above inequality is since $\log\left(\frac{\det\left(X_1^{-1}\right)}{\det\left(X_0^{-1}\right)}\right) \geq 0$ using Equation (27) $\qquad\square$

**Lemma 10** (Upperbound of $T_2$).

$$T_2 \leq \frac{1}{2\eta} \cdot (CT + \epsilon D_2^2)\left(\frac{1+\beta}{1-\beta}\right) \cdot \sqrt{\frac{2}{3} \cdot T \cdot \left(\sum_{i=1}^{n} \log\left(1 + \frac{\left\| g_{1:T}^{(i)} \right\|_2^2}{\epsilon}\right) + \sum_{i=1}^{n-1} \log\left(1 - \beta_i^2\right)\right)}$$

*Proof.* The proof directly follows from Equation (25) and Lemma 9 ∎

**Lemma 11** (Upperbound of $T_3$). *Let* $X_0 = \arg\min_{X \in S_n(\mathcal{G})^{++}} D_{\ell d}(X, H_0^{-1})$

$$
\begin{aligned}
T_3 &= \sum_{t=1}^{T} \frac{\eta}{2} \cdot g_t^T X_t g_t \\
&\leq \frac{\eta}{2} \cdot \log\left(\frac{\det\left(X_T^{-1}\right)}{\det\left(X_0^{-1}\right)}\right) \\
&\leq \frac{\eta}{2} \cdot \sum_{i=1}^{n} \log\left(1 + \frac{\left\|g_{1:T}^{(i)}\right\|_2^2}{\epsilon}\right) + \sum_{i=1}^{n-1} \log\left(1 - \beta_i^2\right)
\end{aligned}
$$

*where* $\beta_i = \dfrac{((g_{1:T}^{(i)})^T g_{1:T}^{(i+1)})^2}{(\epsilon + \left\|g_{1:T}^{(i)}\right\|_2^2) \cdot (\epsilon + \left\|g_{1:T}^{(i+1)}\right\|_2^2)}$

*Proof.*

$$
\begin{aligned}
g_t^T X_t g_t &= \text{Tr}(X_t g_t g_t^T) \\
&= \text{Tr}(X_t(X_t^{-1} - X_{t-1}^{-1})) \\
&= n - \text{Tr}(X_t X_{t-1}^{-1}) \\
&\leq -\log\left(\frac{\det\left(X_{t-1}^{-1}\right)}{\det\left(X_t^{-1}\right)}\right)
\end{aligned}
$$

The second equality is using Lemma 3 and that $X_t$'s sparsity graph is tridiagonal. The first inequality is using the property $D_{\ell d}(X_{t-1}^{-1}, X_t^{-1}) \geq 0$ of LogDet divergence. Thus summing up and using Equation (28) will give the lemma. ∎

**Putting together $T_1$, $T_2$ and $T_3$ from Lemma 5, Lemma 10 and Lemma 11**

$$
T_1 \leq \frac{(C + \epsilon D_2^2)}{2\eta} \cdot \frac{1+\beta}{1-\beta},
$$

$$
T_2 \leq \frac{1}{2\eta}(CT + \epsilon D_2^2)\left(\frac{1+\beta}{1-\beta}\right)\sqrt{\frac{2}{3}T\left(\sum_{i=1}^{n} \log\left(1 + \frac{\left\|g_{1:T}^{(i)}\right\|_2^2}{\epsilon}\right) + \sum_{i=1}^{n-1} \log\left(1 - \beta_i^2\right)\right)}, \quad (29)
$$

$$
T_3 \leq \frac{\eta}{2} \cdot \sum_{i=1}^{n} \log\left(1 + \frac{\left\|g_{1:T}^{(i)}\right\|_2^2}{\epsilon}\right) + \sum_{i=1}^{n-1} \log\left(1 - \beta_i^2\right) \quad (30)
$$

If we set $\eta = \dfrac{C^{1/2} T^{3/4} \cdot (\frac{1+\beta}{1-\beta})^{1/2}}{\left(\sum_{i=1}^{n} \log\left(1 + \frac{\left\|g_{1:T}^{(i)}\right\|_2^2}{\epsilon}\right) + \sum_{i=1}^{n-1} \log\left(1-\beta_i^2\right)\right)^{1/4}}$, then

$$
\begin{aligned}
R_T &\leq T_1 + T_2 + T_3 \\
&\leq \mathcal{O}\left(C^{1/2} \cdot T^{3/4} \cdot \left(\frac{1+\beta}{1-\beta}\right)^{1/2} \cdot \left(\sum_{i=1}^{n} \log\left(1 + \frac{\left\|g_{1:T}^{(i)}\right\|_2^2}{\epsilon}\right) + \sum_{i=1}^{n-1} \log\left(1 - \beta_i^2\right)\right)^{3/4}\right)
\end{aligned}
$$

### A.3  $\mathcal{O}(T^{1/2})$ REGRET UPPER BOUND

In this section we derive a regret upper bound with a $\mathcal{O}(T^{1/2})$ growth, in contrast to the $\mathcal{O}(T^{3/4})$ obtained in A.2. In (29), $T_2 = \sum_{t=2}^{T}(w_t - w^*)^T(X_t^{-1} - X_{t-1}^{-1})(w_t - w^*)$ is of the order $\mathcal{O}(T^{3/2})$, which can be reduced to $\mathcal{O}(T)$, by upper bounding each entry of $X_t^{-1} - X_{t-1}^{-1}$ individually. The following lemma helps in constructing a telescoping argument to bound $\left|(X_t^{-1} - X_{t-1}^{-1})_{i,j}\right|$.

**Lemma 12.** *Let $H, \tilde{H} \in S_n^{++}$, such that $\tilde{H} = H + gg^T$, where $g \in \mathbb{R}^n$, then*

$$\frac{\tilde{H}_{ij}}{\sqrt{\tilde{H}_{ii}\tilde{H}_{jj}}} - \frac{H_{ij}}{\sqrt{H_{ii}H_{jj}}} = \frac{g_i g_j}{\sqrt{\tilde{H}_{ii}\tilde{H}_{jj}}} + \frac{H_{ij}}{\sqrt{H_{ii}H_{jj}}}\left(\sqrt{\frac{H_{ii}H_{jj}}{\tilde{H}_{ii}\tilde{H}_{jj}}} - 1\right) = \theta_{ij}$$

*Proof.*

$$\frac{\tilde{H}_{ij}}{\sqrt{\tilde{H}_{ii}\tilde{H}_{jj}}} - \frac{H_{ij}}{\sqrt{H_{ii}H_{jj}}} = \frac{1}{\sqrt{H_{ii}H_{jj}}}(\tilde{H}_{ij}\frac{\sqrt{H_{ii}H_{jj}}}{\sqrt{\tilde{H}_{ii}\tilde{H}_{jj}}} - H_{ij})$$

$$= \frac{1}{\sqrt{H_{ii}H_{jj}}}\left(g_i g_j \frac{\sqrt{H_{ii}H_{jj}}}{\sqrt{\tilde{H}_{ii}\tilde{H}_{jj}}} + H_{ij}\left(\frac{\sqrt{H_{ii}H_{jj}}}{\sqrt{\tilde{H}_{ii}\tilde{H}_{jj}}} - 1\right)\right)$$

$\square$

**Lemma 13.** *Let $H, \tilde{H} \in S_n^{++}$, such that $\tilde{H} = H + gg^T$, where $g \in \mathbb{R}^n$. Also, $\tilde{Y} = \arg\min_{X \in S_n(\mathcal{G})^{++}} D_{\ell d}(X, \tilde{H})$ and $Y = \arg\min_{X \in S_n(\mathcal{G})^{++}} D_{\ell d}(X, H)$, where $\mathcal{G}$ is a chain graph, then*

$$\left|(\tilde{Y}^{-1} - Y^{-1})_{ii+k}\right| \leq G_\infty^2 \kappa(k\beta + k + 2)\beta^{k-1},$$

*where $i, i + k \leq n$, $G_\infty = \|g\|_\infty$ and $\max_{i,j} |H_{ij}|/\sqrt{H_{ii}H_{jj}} \leq \beta < 1$. Let $\kappa(\text{diag}(H)) := $ condition number of the diagonal part of $H$, then $\kappa := \max(\kappa(\text{diag}(H)), \kappa(\text{diag}(\tilde{H})))$.*

*Proof.* Using Lemma 3 will give the following:

$$\left|(\tilde{Y}^{-1} - Y^{-1})_{ii+k}\right| = \left|\frac{\tilde{H}_{ii+1}\dots\tilde{H}_{i+k-1i+k}}{\tilde{H}_{i+1i+1}\dots\tilde{H}_{i+ki+k}} - \frac{H_{ii+1}\dots H_{i+k-1i+k}}{H_{i+1i+1}\dots H_{i+ki+k}}\right|$$

$$= \left|\sqrt{\tilde{H}_{ii}}\tilde{N}_{ii+1}\dots\tilde{N}_{i+k-1i+k}\sqrt{\tilde{H}_{i+ki+k}} - \sqrt{H_{ii}}N_{ii+1}\dots N_{i+k-1i+k}\sqrt{H_{i+ki+k}}\right|$$

$$= \sqrt{\tilde{H}_{ii}\tilde{H}_{i+ki+k}}\left|\tilde{N}_{ii+1}\dots\tilde{N}_{i+k-1i+k} - N_{ii+1}\dots N_{i+k-1i+k}\sqrt{\frac{H_{ii}H_{i+ki+k}}{\tilde{H}_{ii}\tilde{H}_{i+ki+k}}}\right|$$

$\square$

where $N_{ij} = H_{ij}/\sqrt{H_{ii}H_{jj}}$. Expanding $\tilde{N}_{ii+1} = N_{ii+1} + \theta_{ii+1}$ (from Lemma 12), subsequently $\tilde{N}_{ii+2} = N_{ii+2} + \theta_{ii+2}$ and so on will give

$$\left| \tilde{N}_{ii+1} \ldots \tilde{N}_{i+k-1i+k} - N_{ii+1} \ldots N_{i+k-1i+k}\sqrt{\frac{H_{ii}H_{i+ki+k}}{\tilde{H}_{ii}\tilde{H}_{i+ki+k}}} \right| =$$

$$\left| \theta_{ii+1}\tilde{N}_{i+1i+2} \ldots \tilde{N}_{i+k-1i+k} + N_{ii+1}\left( \tilde{N}_{i+1i+2} \ldots \tilde{N}_{i+k-1i+k} - N_{i+1i+2} \ldots N_{i+k-1i+k}\sqrt{\frac{H_{ii}H_{i+ki+k}}{\tilde{H}_{ii}\tilde{H}_{i+ki+k}}} \right) \right|$$

$$= |\theta_{ii+1}\tilde{N}_{i+1i+2} \ldots \tilde{N}_{i+k-1i+k} + N_{ii+1}\theta_{i+1i+2}\tilde{N}_{ii+3} \ldots \tilde{N}_{i+k-1i+k} + \ldots + N_{ii+1} \ldots N_{ii+k-1}\theta_{i+k-1i+k}$$

$$+ N_{ii+1} \ldots N_{ii+k}\left( 1 - \sqrt{\frac{H_{ii}H_{i+ki+k}}{\tilde{H}_{ii}\tilde{H}_{i+ki+k}}} \right)|$$

$$\leq (\sum_{l=0}^{k-1} |\theta_{i+li+l+1}|)\beta^{k-1} + \beta^{k-1}\left| 1 - \sqrt{\frac{H_{ii}H_{i+ki+k}}{\tilde{H}_{ii}\tilde{H}_{i+ki+k}}} \right|,$$

$$\implies \left| (\tilde{Y}^{-1} - Y^{-1})_{ii+k} \right| \leq \sqrt{\tilde{H}_{ii}\tilde{H}_{i+ki+k}}\left( (\sum_{l=0}^{k-1} |\theta_{i+li+l+1}|)\beta^{k-1} + \beta^{k-1}\left| 1 - \sqrt{\frac{H_{ii}H_{i+ki+k}}{\tilde{H}_{ii}\tilde{H}_{i+ki+k}}} \right| \right)$$

where $\max_{i,j} |N_{i,j}|, \; \max_{i,j} |\tilde{N}_{i,j}| \leq \beta$. Expanding $\theta_{i+li+l+1}$ from Lemma 12 in the term $|\theta_{i+li+l+1}|\sqrt{\tilde{H}_{ii}\tilde{H}_{i+ki+k}}$ will give:

$$|\theta_{i+li+l+1}|\sqrt{\tilde{H}_{ii}\tilde{H}_{i+ki+k}} = \left| \sqrt{\tilde{H}_{ii}\tilde{H}_{i+ki+k}}\frac{g_{i+l}g_{i+l+1}}{\sqrt{\tilde{H}_{i+li+l}\tilde{H}_{i+l+1i+l+1}}} + \right.$$

$$\left. \sqrt{\tilde{H}_{ii}\tilde{H}_{i+ki+k}}N_{i+li+l+1}\left( \sqrt{\frac{H_{i+li+l}H_{i+l+1i+l+1}}{\tilde{H}_{i+li+l}\tilde{H}_{i+l+1i+l+1}}} - 1 \right) \right|$$

$$\leq \left| \sqrt{\tilde{H}_{ii}\tilde{H}_{i+ki+k}}\frac{g_{i+l}g_{i+l+1}}{\sqrt{\tilde{H}_{i+li+l}\tilde{H}_{i+l+1i+l+1}}} \right| +$$

$$\left| \sqrt{\tilde{H}_{ii}\tilde{H}_{i+ki+k}}N_{i+li+l+1}\left( 1 - \sqrt{\frac{H_{i+li+l}H_{i+l+1i+l+1}}{\tilde{H}_{i+li+l}\tilde{H}_{i+l+1i+l+1}}} \right) \right|$$

Since $H_{i+li+l}H_{i+l+1i+l+1} \leq \tilde{H}_{i+li+l}\tilde{H}_{i+l+1i+l+1}$,

$$1 - \sqrt{\frac{H_{i+li+l}H_{i+l+1i+l+1}}{\tilde{H}_{i+li+l}\tilde{H}_{i+l+1i+l+1}}} \leq \max\left( 1 - \frac{H_{i+li+l}}{\tilde{H}_{i+li+l}}, 1 - \frac{H_{i+l+1i+l+1}}{\tilde{H}_{i+l+1i+l+1}} \right)$$

$$\leq \max\left( \frac{g_{i+l}^2}{\tilde{H}_{i+li+l}}, \frac{g_{i+l+1}^2}{\tilde{H}_{i+l+1i+l+1}} \right)$$

Using the above, $H_{i,i}/H_{j,j} \leq \kappa$, and $|g_i| \leq G_\infty, \forall i,j \in [n]$, gives

$$\sqrt{\tilde{H}_{ii}\tilde{H}_{i+ki+k}}|\theta_{i+li+l+1}| \leq G_\infty^2\kappa + \beta G_\infty^2\kappa$$

$$\leq G_\infty^2\kappa(1 + \beta)$$

Thus the following part of $\left| \left( \tilde{Y}^{-1} - Y^{-1} \right)_{ii+k} \right|$ can be upperbounded:

$$\sqrt{\tilde{H}_{ii}\tilde{H}_{i+ki+k}}\left( (\sum_{l=0}^{k-1} |\theta_{i+li+l+1}|)\beta^{k-1} \right) \leq G_\infty^2\kappa(1 + \beta)k\beta^{k-1}$$

Also, $\beta^{k-1}\left| 1 - \sqrt{\frac{H_{ii}H_{i+ki+k}}{\tilde{H}_{ii}\tilde{H}_{i+ki+k}}} \right| \leq \beta^{k-1}\kappa G_\infty^2$, so

$$\left| \left( \tilde{Y}^{-1} - Y^{-1} \right)_{ii+k} \right| \leq G_\infty^2\kappa(k\beta + k + 2)\beta^{k-1}$$

**Lemma 14** ($\mathcal{O}(T)$ upper bound of $T_2$)**.** *Given that $\kappa(\mathrm{diag}(H_t)) \leq \kappa$, $\|w_t - w^*\|_2 \leq D_2$, $\max_{i,j} |(H_t)_{ij}|/\sqrt{(H_t)_{ii}(H_t)_{jj}} \leq \beta < 1$, $\forall t \in [T]$ in Algorithm 1, then $T_2$ in (29) can be bounded as follows:*

$$T_2 \leq \mathcal{O}\left(\frac{T}{2\eta(1-\beta)^2}(G_\infty D_2)^2\kappa\right)$$

*Proof.* Note that $T_2 = \frac{1}{2\eta} \cdot \sum_{t=1}^{T-1}(w_{t+1} - w^*)^T(X_{t+1}^{-1} - X_t^{-1})(w_{t+1} - w^*) \leq \sum_{t=1}^{T-1} D_2^2 \left\|(X_{t+1}^{-1} - X_t^{-1})\right\|_2$. Using $\|A\|_2 = \rho(A) \leq \|A\|_\infty$ for symmetric matrices $A$, we get

$$\left\|X_{t+1}^{-1} - X_t^{-1}\right\|_2 \leq \|X_{t+1}^{-1} - X_t^{-1}\|_\infty$$
$$= \max_i(\sum_j \left|(X_{t+1}^{-1} - X_t^{-1})_{ij}\right|)$$
$$\leq \mathcal{O}(\frac{G_\infty^2\kappa}{(1-\beta)^2})$$

The third inequality is using Lemma 13. Expanding $T_2$ with this bound gives the result. □

**Theorem 6** ($\mathcal{O}(T^{1/2})$ regret upper bound)**.** *Setting*

$$\eta = \frac{T^{1/2}D_2G_\infty\sqrt{\kappa}}{(1-\beta)\sqrt{\sum_{i=1}^n \log\left(1 + \frac{\left\|g_{1:T}^{(i)}\right\|_2^2}{\epsilon}\right) + \sum_{i=1}^{n-1}\log(1-\beta_i^2)}},$$

*gives the following regret*

$$R_T \leq \mathcal{O}\left(\frac{1}{(1-\beta)}T^{1/2}D_2G_\infty\sqrt{\kappa}\sqrt{\sum_{i=1}^n \log\left(1 + \frac{\left\|g_{1:T}^{(i)}\right\|_2^2}{\epsilon}\right) + \sum_{i=1}^{n-1}\log(1-\beta_i^2)}\right),$$

*where condition number $\kappa(\mathrm{diag}(H_t)) \leq \kappa$, $\|w_t - w^*\|_2 \leq D_2$, $\max_{i,j} |(H_t)_{ij}|/\sqrt{(H_t)_{ii}(H_t)_{jj}} \leq \beta < 1$, $\forall t \in [T]$ and $g_t, H_t$ are as defined in Algorithm 1.*

*Proof.* The result follows from using (30) and Lemma 14. □

### A.4 NUMERICAL STABILITY

**Theorem 7** (*Full version of Theorem 4*)**.** *Let $H \in S_n^{++}$ such that $H_{ii} = 1$, for $i \in [n]$ and a symmetric perturbation $\Delta H$ such that $\Delta H_{ii} = 0$, for $i \in [n]$ and $H + \Delta H \succ 0$. Let $\hat{X} = \arg\min_{X \in S_n(\mathcal{G})^{++}} \mathrm{D}_{\ell d}\left(X, H^{-1}\right)$ and $\hat{X} + \Delta\hat{X} = \arg\min_{X \in S_n(\mathcal{G})^{++}} \mathrm{D}_{\ell d}\left(X, (H + \Delta H)^{-1}\right)$, here $\mathcal{G} :=$ chain/tridiagonal sparsity graph and $S_n(\mathcal{G})^{++}$ denotes positive definite matrices which follows the sparsity pattern $\mathcal{G}$.*

$$\kappa_{\ell d} = \max_{|i-j|\leq 1} \limsup_{\epsilon \to 0} \left\{\frac{\left|\Delta\hat{X}_{ij}\right|}{\epsilon\left|\hat{X}_{ij}\right|} : |\Delta H_{k,l}| \leq |\epsilon H_{k,l}|, (k,l) \in E_\mathcal{G}\right\}$$
$$\leq \max_{i\in[n-1]} 1/(1-\beta_i^2)$$

*where, $\kappa_{\ell d} :=$ condition number of the LogDet subproblem, $\kappa_2(.) :=$ condition number of a matrix in $\ell_2$ norm, $\beta_i = H_{ii+1}/\sqrt{H_{ii}H_{i+1i+1}}$*

*Proof.* Consider the offdiagonals for which $(\hat{X} + \Delta\hat{X})_{ii+1} = -H_{ii+1}/(1 - H_{ii+1}^2) = f(H_{ii+1})$, where $f(x) = -x/(1-x^2)$. Let $y = f(x)$, $\hat{y} = f(x + \Delta x)$ and $|\Delta x/x| \leq \epsilon$ then

using Taylor series

$$\left|\frac{(\hat{y}-y)}{y}\right| = \left|\frac{xf'(x)}{f(x)}\right|\left|\frac{\Delta x}{x}\right| + O((\Delta x)^2)$$

$$\implies \lim_{\epsilon\to 0}\left|\frac{(\hat{y}-y)}{\epsilon y}\right| \leq \frac{xf'(x)}{f(x)}$$

Using the above inequality, with $x := H_{ii+1}$ and $y := \hat{X}_{ii+1}$,

$$\lim_{\epsilon\to 0}\left|\frac{\Delta \hat{X}_{ii+1}}{\epsilon \hat{X}_{ii+1}}\right| \leq \frac{1+H_{ii+1}^2}{1-H_{ii+1}^2} \tag{31}$$

$$\leq \frac{2}{1-H_{ii+1}^2}$$

Let $g(x) = x^2/(1-x^2)$, let $y_1 = g(w_1)$, $y_2 = g(x_2)$, $\hat{y}_1 = g(w_1+\Delta x)$, $\hat{y}_2 = g(x_2+\Delta x)$. Using Taylor series

$$\left|\frac{(\hat{y}_1-y_1)}{y_1}\right| = \left|\frac{x_1 f'(x_1)}{f(x_1)}\right|\left|\frac{\Delta x_1}{x_1}\right| + O((\Delta x_1)^2)$$

$$\left|\frac{(\hat{y}_2-y_2)}{y_2}\right| = \left|\frac{x_2 f'(x_2)}{f(x_2)}\right|\left|\frac{\Delta x_2}{x_2}\right| + O((\Delta x_2)^2)$$

$$\implies \lim_{\epsilon\to 0}\frac{\Delta y_1 + \Delta y_2}{\epsilon(1+y_1+y_2)} \leq \max\left(\frac{2}{1-x_1^2}, \frac{2}{1-x_2^2}\right)$$

Putting $x_1 := H_{ii+1}$, $x_2 := H_{ii-1}$ and analyzing $y_1 := H_{ii+1}^2/(1-H_{ii+1}^2)$ and $y_2 := H_{ii-1}^2/(1-H_{ii-1}^2)$ will result in the following

$$\lim_{\epsilon\to 0}\left|\frac{\Delta \hat{X}_{ii}}{\hat{X}_{ii}}\right| \leq \max\left(\frac{2}{1-H_{ii+1}^2}, \frac{2}{1-H_{ii-1}^2}\right) \tag{32}$$

Since $\hat{X}_{ii} = 1 + H_{ii+1}^2/(1-H_{ii+1}^2) + H_{ii-1}^2/(1-H_{ii-1}^2)$. Putting together Equation (32) and Equation (31), the theorem is proved. $\qquad\square$

*Proof of Lemma 1*

For $b = 1$, if $g_{1:T}^{(j)} = g_{1:T}^{(j+1)}$, then $H_{jj+1} = H_{jj} = H_{j+1j+1} = \left\|g_{1:T}^{(j)}\right\|_2^2$, thus $H_{jj} - H_{jj+1}^2/H_{j+1j+1} = 0$.

For $b > 1$, since $H_{I_j I_j}$, using Guttman rank additivity formula, $\mathrm{rank}(H_{jj} - H_{jj+1}^2/H_{j+1j+1}) = \mathrm{rank}(H_{J_j J_j}) - rank(H_{I_j I_j}) = 0$, thus $H_{jj} - H_{jj+1}^2/H_{j+1j+1} = 0$.

Furthermore, if $\mathrm{rank}(H) \leq b$, then all $b+1 \times b+1$ principal submatrices of $H$ have rank $b$, thus $\forall j$, $H_{J_j J_j}$ have a rank $b$, thus $D_{jj}$ for all $j$ are undefined.

*Proof of Theorem 5*

Let $I_i = \{j : i < j, (i,j) \in E_{\mathcal{G}}\}$ and $I_i' = \{j : i < j, (i,j) \in E_{\tilde{\mathcal{G}}}\}$ Let $K = \{i : H_{ii} - H_{I_i i}^T H_{I_i I_i}^{-1} H_{I_i i}$ is undefined or $0, i \in [n]\}$ denote vertices which are getting removed by the algorithm, then for the new graph $\tilde{\mathcal{G}}$, $D_{ii} = 1/H_{ii}, \forall i \in K$ since $H_{ii} > 0$.

Let $\bar{K} = \{i : H_{ii} - H_{I_i i}^T H_{I_i I_i}^{-1} H_{I_i i} > 0, i \in [n]\}$. Let for some $j \in \bar{K}$, if

$$l = \arg\min\{i : j < i, i \in K \cap I_j\},$$

denotes the nearest connected vertex higher than $j$ for which $D_{ll}$ is undefined or zero, then according to the definition $E_{\tilde{\mathcal{G}}}$ in Algorithm 3, $I_j' = \{j+1, \ldots l-1\} \subset I_j$, since $D_{jj}$ is well-defined, $H_{I_j I_j}$ is invertible, which makes it a positive definite matrix (since $H$ is PSD). Since $H_{jj} - H_{I_j j}^T H_{I_j I_j}^{-1} H_{I_j j} > 0$, using Guttman rank additivity formula $H_{J_j J_j} \succ 0$, where $J_j = I_j \cup j$. Since $H_{J_j' J_j'}$ is a submatrix of $H_{J_j J_j}$, it is positive definite and hence its schur complement $H_{jj} - H_{I_j' j}^T H_{I_j' I_j'}^{-1} H_{I_j' j} > 0$. Thus for all $j \in [n]$, the corresponding $D_{jj}$'s are well-defined in the new graph $\tilde{\mathcal{G}}$.

Note that $\kappa_{\ell d}^{\tilde{\mathcal{G}}} = \max_{i\in[n-1]} 1/(1-\beta_i^2) < \max_{i\in\bar{K}} 1/(1-\beta_i^2) = \kappa_{\ell d}^{\mathcal{G}}$, for tridiagonal graph, where $\beta_i = H_{ii+1}$, in the case where $H_{ii} = 1$. This is because the $\arg\max_{i\in[n-1]} 1/(1-\beta_i^2) \in K$.

Table 3: **float32 experiments on Autoencoder benchmark using different band sizes.** Band size 0 corresponds to diag-SONew and 1 corresponds to tridiag-SONew. We see the training loss getting better as we increase band size

| Band size | 0 (diag-SONew) | 1 (tridiag-SONew) | 4 | 10 |
|---|---|---|---|---|
| Train CE loss | 53.025 | 51.723 | 51.357 | 51.226 |

Table 4: **bfloat16 experiments on Autoencoder benchmark with and without Algorithm 3.** We observe gains in training loss when using Algorithm 3

| Optimizer | Train CE loss - without Algorithm 3 | Train CE loss - with Algorithm 3 |
|---|---|---|
| tridiag-SONew | 53.150 | 51.936 |
| band-4-SONew | 51.950 | 51.84 |

Table 5: **Large Scale Benchmarks.** We compare tds vs Adam on the following large scale benchmark. We compare train CE loss, and validation performance - measured as precision for OGBG benchmark and error rate for Resnet50 and Vision Transformer benchmark.

| Benchmark | # model parameters | # training points | Train Loss (Adam) | Train Loss(SONew) | Valid. perf. (Adam) | Valid. perf. (SONew) |
|---|---|---|---|---|---|---|
| Resnet50-Imagenet | 204M | 1.2M | 0.0951 | **0.0857** | 22.61% | **22.55%** |
| Vision Tansformer-Imagenet | 176M | 1.2M | 0.4589 | **0.45506** | **22.85%** | 23.30% |
| OGBG-molpcba | 28.4M | 437,929 | **0.0145** | 0.0157 | **0.2835** | 0.2820 |

## A.5 ADDITIONAL EXPERIMENTS, ABLATIONS, AND DETAILS

**Effect of band size in banded-SONew** Increasing band size in banded-SONew captures more correlation between parameters, hence should expectedly lead to better preconditioners. We confirm this through experiments on the Autoencoder benchmark where we take band size = 0 (diag-SONew), 1 (tridiag-SONew), 4, and 10.

**Effect of mini-batch size** To find the effect of mini-batch size, in Table 7, We empirically compare SONew with state of the art first-order methods such as Adam and RMSProp, and second-order method Shampoo. We see that SONew performance doesn't deteriorate much when using smaller or larger batch size. First order methods on the other hand suffer significantly. We also notice that Shampoo doesn't perform better than SONew in these regimes.

**Effect of Numerical Stability Algorithm 3** On tridiag-SONew and banded-4-SONew, we observe that using Algorithm 3 improves training loss. We present in Table 4 results where we observed significant performance improvements.

**Large Scale Benchmark Comparison** To test efficacy of SONew in Deep Learning, we test our method against Adam on 3 large scale benchmarks - Resnet50 He et al. (2015b) on Imagenet Deng et al. (2009) training, Vision Transformer Dosovitskiy et al. (2020) on Imagenet training, and Graph-Network Battaglia et al. (2018); Godwin* et al. (2020) on OGBG-molpcba dataset Hu et al. (2020). The numbers are given in Table 5. We see that SONew performs comparable or outperforms Adam in all the benchmark in both train loss and validation performance. Moreover, we plot the training and validation curve in Figure 2. We observe that SONew has an early advantage over Adam.

**Hyperparaeter search space** We provide the hyperparamter search space for experiments presented in Section 6. We search over $2k$ hyperparameters for each experiment using a Bayesian Optimization package. The search ranges are: first order momentum term $\beta_1 \in [1e-1, 0.999]$, second order momentum term $\beta_2 \in [1e-1, 0.999]$, learning rate $\in [1e-7, 1e-1]$, $\epsilon \in [1e-10, 1e-1]$. We give the optimal hyperparameter value for each experiment in Table 6. For large scale benchmark (Table 5), we search $\beta_1, \beta_2 \in [0.1, 0.999]$, $lr \in [1e-5, 1e-1]$, $\epsilon \in [1e-8, 1e-4]$, weight decay $\in [1e-5, 1.0]$, learning rate warmup $\in [2\%, 5\%, 10\%]*total\_train\_steps$. We use cosine learning rate schedule. For resnet50 imagenet, we also search label smoothing over $[0.0, 0.2]$. Batch size was kept = 1024, 1024, and 512 for Resnet50, Vision Transformer, and OGBG respectively. We sweep over 100 hyperparameters in the search space for both SONew and Adam.

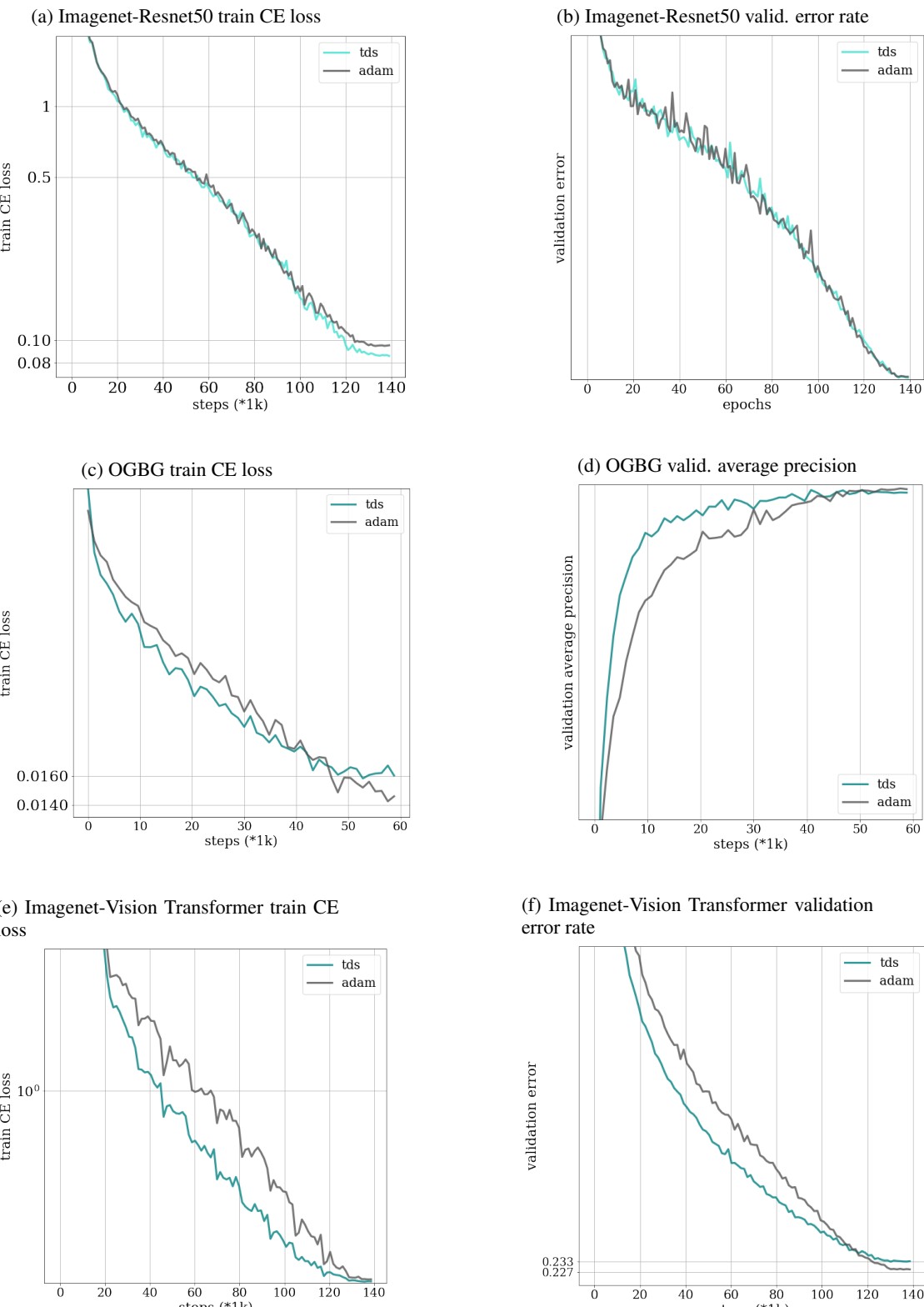

Figure 2: SONew vs Adam on large scale benchmark - Train Cross Entropy (CE) Loss (left column) and validation performance (right column). We see that SONew has an early advantage over Adam in all the experiments.

Table 6: **Optimal hyperparamers for Autoencoder Benchmark**

(a) float32 experiments optimal hyperparamters

| Baseline | $\beta_1$ | $\beta_2$ | $\epsilon$ | lr |
|---|---|---|---|---|
| SGD | 0.99 | 0.91 | 8.37e-9 | 1.17e-2 |
| Nesterov | 0.914 | 0.90 | 3.88e-10 | 5.74e-3 |
| Adagrad | 0.95 | 0.90 | 9.96e-7 | 1.82e-2 |
| Momentum | 0.9 | 0.99 | 1e-5 | 6.89e-3 |
| RMSProp | 0.9 | 0.9 | 1e-10 | 4.61e-4 |
| Adam | 0.9 | 0.94 | 1.65e-6 | 3.75e-3 |
| Diag-SONew | 0.88 | 0.95 | 4.63e-6 | 1.18e-3 |
| Shampoo | 0.9 | 0.95 | 9.6e-9 | 3.70e-3 |
| tridiag | 0.9 | 0.96 | 1.3e-6 | 8.60e-3 |
| band-4 | 0.88 | 0.95 | 1.5e-3 | 5.53e-3 |

(b) bfloat16 experiments optimal hyperparamters

| Baseline | $\beta_1$ | $\beta_2$ | $\epsilon$ | lr |
|---|---|---|---|---|
| SGD | 0.96 | 0.98 | 2.80e-2 | 1.35e-2 |
| Nesterov | 0.914 | 0.945 | 8.48e-9 | 6.19e-3 |
| Adagrad | 0.95 | 0.93 | 2.44e-5 | 2.53e-2 |
| Momentum | 0.9 | 0.99 | 0.1 | 7.77e-3 |
| RMSProp | 0.9 | 0.9 | 2.53e-10 | 4.83e-4 |
| Adam | 0.9 | 0.94 | 3.03e-10 | 3.45e-3 |
| Diag-SONew | 0.9 | 0.95 | 4.07e-6 | 8.50e-3 |
| Shampoo | 0.85 | 0.806 | 6.58e-4 | 5.03e-3 |
| ztridiag | 0.83 | 0.954 | 1.78e-6 | 7.83e-3 |
| band-4 | 0.9 | 0.96 | 1.52e-6 | 4.53e-3 |

Table 7: **Comparison on Autoencoder with different batch-sizes**

| Baseline\Batch size | 100 | 1000 | 5000 | 10000 |
|---|---|---|---|---|
| RMSProp | 55.61 | 53.33 | 58.69 | 64.91 |
| Adam | 55.67 | 54.39 | 58.93 | 65.37 |
| Shampoo(20) | 53.91 | 50.70 | 53.52 | 54.90 |
| tds | 53.84 | 51.72 | 54.24 | 55.87 |
| bds-4 | 53.52 | 51.35 | 53.03 | 54.89 |

