# OpenReview forum: "A Computationally Efficient Sparsified Online Newton Method"
_ICLR.cc/2023/Conference — Submitted to ICLR 2023_

### Official Review · Reviewer_9vTP · 2022-10-20

**Confidence:** 3
**Correctness:** 4
**Technical Novelty And Significance:** 2
**Empirical Novelty And Significance:** 2
**Recommendation:** 3

**Clarity, Quality, Novelty And Reproducibility:**

This paper is well-written and easy to follow. However, the authors missed some very related work [1] [2]. Moreover, although the use of $b$-banded sparsity graph to accelerate ONS is new to me, I am not convinced that it is better than existing efficient variants of ONS.

[1] Haipeng Luo et al. Efficient Second Order Online Learning via Sketching. In NeurIPS 2016.

[2] Luo Luo et al. Robust Frequent Directions with Application in Online Learning. In JMLR 2019.


**Strength And Weaknesses:**

#Strength
1) The authors proposed a new efficient variant of online Newton step (ONS) by using $b$-banded sparsity graph.
2) The proposed algorithm improves the time complexity of ONS from $O(n^2)$ to $O(b^3 n)$.

#Weaknesses
1) The authors only established a regret bound of $O(T^{3/4})$ for their method, which is far worse than the $O(\sqrt{T})$ regret bound of many existing first-order methods such as online gradient descent. Note that these first-order methods are more efficient than the proposed algorithm for $b>1$.
2) The experiments presented in this paper is too simple, which is not sufficient to verify the advantage of the proposed method. The authors at least consider some other datasets such as CIFAR10 and CIFAR100. Moreover, the authors should also consider other deep neural networks such as convolutional neural networks (CNNs).
3) In the experiments, some tricks including momentum and grafting are added to the proposed method, which may affect the evaluation of the proposed method.
4) There actually exist efficient variants of ONS [1][2], which enjoy the time complexity of $O(\tau n)$ for very small $\tau$. However, the authors do not discuss and compare the proposed method with them.

[1] Haipeng Luo et al. Efficient Second Order Online Learning via Sketching. In NeurIPS 2016.

[2] Luo Luo et al. Robust Frequent Directions with Application in Online Learning. In JMLR 2019.


**Summary Of The Paper:**

This paper proposed SONew (Sparsified Online Newton), which is an efficient variant of online Newton step (ONS) by using $b$-banded sparsity graph. The time complexity of this method is $O(b^3 n)$, and the regret bound is $O(T^{3/4})$. The authors provide some simple experiments to verify the performance of the proposed method.

**Summary Of The Review:**

By considering the $O(T^{3/4})$ regret bound of the proposed method and the existing efficient variants of ONS, I tend to reject this paper.

---

> ### Author Response · Authors · 2022-11-18
> **Response to Reviewer 3**
>
> Thank you for your comments.
>
> >The authors only established a $\mathcal{O}(T^{3/4})$ regret bound of for their method, which is far worse than the $\mathcal{O}(T^{1/2})$ regret bound of many existing first-order methods such as online gradient descent. Note that these first-order methods are more efficient than the proposed algorithm for .
>
> Following your suggestion, we conducted more analysis to reduce growth of $T$ in our regret bound to $\mathcal{O}(T^{1/2})$. We added this in Appendix A.3. This analysis used a different approach to bound the second term $T_2 = \sum_{t=2}^{T}(w_{t} - w^*)^T(X_{t}^{-1}-X_{t-1}^{-1})(w_{t} - w^*)$ in (28), by upper bounding the entries of $X_t^{-1}-X_{t-1}^{-1}$ individually by a constant. In Theorem 6 we establish a regret upper bound of $\mathcal{O}(T^{1/2})$ given that matrices $\text{diag}(H_t)$ are well conditioned.
>
> >The experiments presented in this paper is too simple, which is not sufficient to verify the advantage of the proposed method. The authors at least consider some other datasets such as CIFAR10 and CIFAR100. Moreover, the authors should also consider other deep neural networks such as convolutional neural networks (CNNs).
>
> Thanks for the suggestion. We ran SONew and compared it to Adam over 3 large scale benchmarks - Resnet50 on Imagenet, Vision Transformer on Imagenet, and GNN on OGBG-MOLPCBA dataset. We added the training and validation plots in Appendix A.5 and observed that SONew performs comparably or outperforms Adam while having an early advantage in convergence on almost every experiment. This exhibits the efficacy of SONew and its fast convergence in the deep learning setting.
>
> >In the experiments, some tricks including momentum and grafting are added to the proposed method, which may affect the evaluation of the proposed method.
>
> Momentum is a commonly used technique to train deep-learning models [5,6]. In our experiments we used this technique fairly by offering an equal search space window to the momentum hyperparameter $\beta_1\in(0.1,0.999)$ for both baselines (Shampoo and first order methods) and SONew.
> Grafting is commonly used to improve performance of second-order methods [4], where the direction of the update of the second order method is retained, and the magnitude of a first order method is used instead. This technique is used for both Shampoo (our second-order baseline) and SONew for a fair comparison. Note that using grafting on first order methods doesn’t result in a better performance than the best first order method [3].
>
> >There actually exist efficient variants of ONS [1,2] which enjoy the time complexity of $\mathcal{O}(\tau n)$ for very small $\tau$. However, the authors do not discuss and compare the proposed method with them
>
> Thanks for pointing out these papers. We added the following comparison to the Related Work section.
> "There is prior work [1,2] in reducing the complexity - $\mathcal{O}(n^2)$ flops  of Online Newton Step (ONS) to $\mathcal{O}(n)$ flops using sketching. These ONS variants maintain a low rank approximation of $H_t$ (as in Algorithm 1) and updating it with a new gradient $g_t$ at every iteration requires conducting SVD [2]/orthonormalization [1] of a tall and thin matrix in $\mathbb{R}^{n\times r}$, where $r$ denotes the rank of approximation of $H_t$. Our proposed method (Algorithm 1) has a more parallelizable update $H_t\coloneqq H_{t-1} + P_{\mathcal{G}}(g_t g_t^T)$ and is more suitable for DNN training."
>
> [1] Haipeng Luo et al. Efficient Second Order Online Learning via Sketching. In NeurIPS 2016.
>
> [2] Luo Luo et al. Robust Frequent Directions with Application in Online Learning. In JMLR 2019.
>
> [3] Naman Agarwal, Rohan Anil, Elad Hazan, Tomer Koren, and Cyril Zhang. Learning rate grafting: Transferability of optimizer tuning, 2022.
>
> [4] Rohan Anil, Vineet Gupta, Tomer Koren, Kevin Regan, and Yoram Singer. Scalable second order optimization for deep learning. arXiv preprint arXiv:2002.09018, 2020.
>
> [5] Vineet Gupta, Tomer Koren, and Yoram Singer. Shampoo: Preconditioned stochastic tensor optimization. In International Conference on Machine Learning, pages 1842–1850. PMLR, 2018
>
> [6]  Diederik P Kingma and Jimmy Ba. Adam: A method for stochastic optimization. arXiv preprint arXiv:1412.6980, 2014

---

> > ### Comment · Reviewer_9vTP · 2022-12-10
> > **Additional Comments**
> >
> > Thank the authors for the response. However, my concern about the related work [1, 2] is not addressed. The authors should also compare the existing variant of ONS by conducting some experiments and discussing the theoretical guarantees.

---

### Official Review · Reviewer_pCLb · 2022-10-24

**Confidence:** 3
**Correctness:** 4
**Technical Novelty And Significance:** 4
**Empirical Novelty And Significance:** 1
**Recommendation:** 5

**Clarity, Quality, Novelty And Reproducibility:**

This paper clearly describes the proposed method's background, motivation, derivation, and significance. It provides a novel combination of LogDet divergence measure and structured sparsity in preconditioning matrix in online convex optimization settings. The quality of the empirical experiments needs to be improved to assess whether SONew is effective in more realistic deep-learning tasks than a simple autoencoder training task.


**Strength And Weaknesses:**

Strengths
- This paper is clearly written.
- This paper highlights an interesting connection between the regret minimization in OCO and LogDet divergence of the preconditioning matrix, giving a novel insight into the online gradient descent and online Newton method.
- By taking advantage of structural sparsity in the preconditioning matrix, the proposed SONew method gives a way to solve the LogDet divergence subproblem fast enough (enough to be applied every step during training). It seems practical even with a large DNN model.
- Also, thanks to the flexibility of the sparse structure in SONew, the form of the preconditioning matrix (diag, tri-diag, banded) can be determined according to the computational budget, and a straightforward trade-off between convergence speed and computation time is observed in the auto-encoder training experiments in MNIST.

Although SONew is based on solid mathematical intuition and the experimental results show good convergence and numerical stability, the empirical results are not strong enough to support SONew’s effectiveness as a ‘deep learning optimizer’, as described below.

Weaknesses
- Comparisons at different mini-batch sizes have not been made.
    - In mini-batch training, g_t in Equation 5 corresponds to the mini-batch gradient, and X_t^{-1} is known as the (mini-batch) empirical Fisher. It has been pointed out that the mini-batch empirical Fisher loses valuable second-order information for optimization as the mini-batch size increases (Equation 4 in https://www.cs.toronto.edu/~rgrosse/courses/csc2541_2022/readings/L05_normalization.pdf).
    - Therefore, comparisons with different (larger or smaller) mini-batch sizes are desirable, as differences in mini-batch size are expected to affect the convergence of SONew (and the optimization methods being compared). In particular, since large batches are desirable for large-scale training that takes advantage of data parallelism and distributed accelerators (e.g., https://arxiv.org/abs/1802.09941), how well SONew maintains convergence in large-batch training is useful information for practical (esp. large-scale) uses.
- Comparison of numerical stability needs improvement.
    - The numerical stability of the eigenvalue decomposition required by Shampoo depends largely on the condition number of the matrix, which can be significantly improved by the damping value to be added to the diagonal of the matrix. For example, in the experimental code of Anil et al. (2020) (https://arxiv.org/abs/2002.09018), the default value of damping is set to 1e-4 (https://github.com/jettify/pytorch-optimizer/blob/master/torch_optimizer/shampoo.py#L50) to improve the numerical stability. Similarly, large damping values are often used in K-FAC in practice (e.g., 1e-3 in https://github.com/gpauloski/kfac-pytorch/blob/main/kfac/preconditioner.py#L57). Therefore, it is desirable to use Shampoo with a large damping value as a baseline result in the bfloat16 setting.
    - K-FAC can also be implemented using Cholesky decomposition. In addition, there are second-order optimization methods that can take advantage of the low-rank nature of matrices and avoid explicit inverse matrix and eigenvalue decomposition calculations (e.g., SKFAC https://openaccess.thecvf.com/content/CVPR2021/papers/Tang_SKFAC_Training_Neural_Networks_With_Faster_Kronecker-Factored_Approximate_Curvature_CVPR_2021_paper.pdf, SENG https://link.springer.com/article/10.1007/s10915-022-01911-x). Since these methods can be expected to have more robust numerical stability in low-precision training, using only Shampoo (which is less numerically stable) as a baseline for low-precision second-order optimization seems inappropriate for evaluating the relative numerical stability of SONew.
- It is unclear whether SONew is effective as a ‘deep learning optimizer’.
    - From the derivation, SONew is an online 'convex' optimizer. However, since the loss function in deep learning is nonconvex, it is unclear how effective regret minimization achieved by SONew is in minimizing loss. Since the MNIST autoencoder training is a very simple task by today's deep learning standards, training results in a larger, more realistic setting (e.g., training CNNs on ImageNet classification, Transformers on language tasks with cross-entropy loss) would be desirable.
    - Although only training loss is shown in the experiment, I believe that the evaluation of generalization performance is necessary to measure the performance of SONew as a deep learning optimizer. For example, not only the training convergence speed of the optimizer but also its compatibility with regularization, such as weight decay, is an essential subject of discussion (e.g., Adam vs. AdamW https://arxiv.org/abs/1711.05101).

Questions
- Why is there no result for “band-4” with bfloat16 in Figure1 and Table2?
- What are the damping values (espsilon) used for SONew, Shampoo, and other adaptive gradient methods?


**Summary Of The Paper:**

This work proposes SONew, an online convex optimizer derived by combining the regret minimization based on the LogDet divergence measure and structured sparsity in the preconditioning matrix. The preconditioning matrix of the SONew is a shape of either diagonal, tri-diagonal, or banded and is calculated in linear flops and numerically stable ways, making it compatible with large deep neural networks. On the MNIST autoencoder training task, SONew achieved better convergence than the existing first-order adaptive gradient method. Furthermore, SONew shows better numerical stability than other methods, including the Shampoo optimizer, when training with bfloat16.


**Summary Of The Review:**

This work proposes a compute- and memory-efficient online convex optimizer for deep neural networks based on solid mathematical insight. However, it is not clear from the given experimental results and theoretical justification alone that the proposed method is helpful in “Learning Representation.” I think more empirical results and discussion of non-convex optimization/generalization would improve the quality of this research, but it is currently the under the bar of acceptance.

---

> ### Author Response · Authors · 2022-11-19
> **Response to Review 2**
>
>
> Thank you for such detailed review on our paper! We conducted a few experiments to address the weaknesses mentioned:
>
> >Therefore, comparisons with different (larger or smaller) mini-batch sizes are desirable, as differences in mini-batch size are expected to affect the convergence of SONew (and the optimization methods being compared). In particular, since large batches are desirable for large-scale training that takes advantage of data parallelism and distributed accelerators (e.g., https://arxiv.org/abs/1802.09941), how well SONew maintains convergence in large-batch training is useful information for practical (esp. large-scale) uses.
>
> To find the effect of batch-size, we empirically compare SONew with state of the art first-order methods like Adam and RMSProp, and second-order method Shampoo. We see that SONew performance doesn't deteriorate much when using smaller or larger batch size. First order methods on the other hand suffer significantly. We also notice that shampoo doesn't perform better than SONew in these regimes.
>
> | Baseline\Batch size |  100  |  1000 | 5000  | 10000 |
> |---------------------|-------|-------|-------|-------|
> | RMSProp             | 55.61 | 53.33 | 58.69 | 64.91 |
> | Adam                | 55.67 | 54.39 | 58.93 | 65.37 |
> | Shampoo(20)         | 53.91 | 50.70 | 53.52 | 54.90 |
> | tds                 | 53.84 | 51.72 | 54.24 | 55.87 |
> | bds-4               | 53.52 | 51.35 | 53.03 | 54.89 |
>
> **Response to Weakness 2, part 1:** We tune all baselines including Shampoo with the damping factor range $\epsilon\in[1e-10, 1e-1]$. Hence, the reported numbers are the best among the entire hyperparams search space that Shampoo could find, which includes large damping values as well. We do a very thorough tuning by searching over 2k hyperparameters for each experiment using a Bayesian Optimization package. We have now included the optimal hyperparams value in Table 6  Appendix A.5.
>
> **Response to Weakness 2, part 2:** We tried running KFAC or SENG but could not make the code run successfully in the limited period we had. We'll surely work on this and include these additional baselines in the final draft.
>
> >Since the MNIST autoencoder training is a very simple task by today's deep learning standards, training results in a larger, more realistic setting (e.g., training CNNs on ImageNet classification, Transformers on language tasks with cross-entropy loss) would be desirable.
>
> Based on your suggestion, we run SONew and compare it with AdamW on 3 large scale benchmarks -  Resnet50 [1] on Imagenet  training, Vision Transformer [2] on Imagenet training, and GraphNetwork [3,4] on OGBG-molpcba dataset [5]. We have added the training and validation plots in the appendix (experiment section) and observe that SONew performs comparably or outperforms Adam while having an early advantage in almost every experiment. This exhibits the efficacy of SONew and its fast convergence in the deep learning setting. We added the following to the introduction of the paper: "We also conduct experiments on large-scale benchmarks in Appendix A.5, and observe comparable or improved performance than Adam [6]". Also, a reference to Appendix A.5 is added in experiments section.
>
> > Why is there no result for “band-4” with bfloat16 in Figure1 and Table2?
>
>  We have now included this in the paper, please revisit Figure 1 and Table 2.
>
> > What are the damping values (espsilon) used for SONew, Shampoo, and other adaptive gradient methods?
>
> We added the optimal hyperparams values on Autoencoder benchmark in Table 6 Appendix A.5.
>
> [1] Kaiming He, Xiangyu Zhang, Shaoqing Ren, and Jian Sun. Deep residual learning for image recognition, 2015.
>
> [2] Jia Deng, Wei Dong, Richard Socher, Li-Jia Li, Kai Li, and Li Fei-Fei. Imagenet: A large-scale hierarchical image database. In 2009 IEEE Conference on Computer Vision and Pattern Recognition, pages 248–255, 2009.
>
> [3] Peter W. Battaglia et al. Relational inductive biases, deep learning, and graph networks, 2018
>
> [4] Jonathan Godwin et al. Jraph: A library for graph neural networks in jax., 2020.
>
> [5] Weihua Hu et al. Open graph benchmark: Datasets for machine learning on graphs, 2020.
>
> [6]  Diederik P Kingma and Jimmy Ba. Adam: A method for stochastic optimization. arXiv preprint arXiv:1412.6980, 2014

---

### Official Review · Reviewer_kEgN · 2022-10-25

**Confidence:** 3
**Clarity, Quality, Novelty And Reproducibility:** This paper is easy to follow but seem…
**Correctness:** 2
**Technical Novelty And Significance:** 2
**Empirical Novelty And Significance:** 2
**Recommendation:** 5

**Strength And Weaknesses:**

Pros: It is easy to follow and the intuition of the algorithm is easy to understand.

Cons: Page 4, between Eq. 8 and Eq. 9, the optimality condition of Eq. 7 is not the gradient equal to zero because Eq. 7 is a constrained optimization problem. Thus, the derivation of Eq. 9 may be problematic.

This paper only considers the third term of Eq.3. This may be not enough because $X_t ^{-1}-X_{t-1}^{-1} $ which makes the regret large.


**Summary Of The Paper:**

This paper proposes sparsifed online Newton method.


**Summary Of The Review:**

This paper proposes sparsifed online Newton method.

---

> ### Author Response · Authors · 2022-11-18
> **Response to Reviewer 1**
>
>
> >Cons: Page 4, between Eq. 8 and Eq. 9, the optimality condition of Eq. 7 is not the gradient equal to zero because Eq. 7 is a constrained optimization problem. Thus, the derivation of Eq. 9 may be problematic.
>
> Thank you for pointing out this error. We fixed this in the revision, by replacing $\nabla (-\log \det(X) + Tr(XP_{\mathcal{G}}(X_{t-1}^{-1}+ g_t g_t^T/\lambda_t)))=0$ with $P_{\mathcal{G}}(\nabla g(X))=0$, where $g(X) = -\log \det(X) + Tr(XP_{\mathcal{G}}(X_{t-1}^{-1}+ g_t g_t^T/\lambda_t))$, and thus $P_{\mathcal{G}}(X_t^{-1}  - X_{t-1}^{-1} - g_t g_t^T/\lambda_t) = 0$, which gives (9) (now (10)). So the derivation is not affected by this error.
>
> >This paper only considers the third term of (3). This may be not enough because $X_t^{-1} - X_{t}^{-1}$ which makes the regret large
>
> In the  objective (4): $$X_t = \arg\min_{X \in S_n^{++}} g_t^T X g_t,\ \ {\text{such that}}\ \ \ D_{\ell d}{(X,X_{t-1})} \le c_t,$$ $g_t^T X g_t$ is minimized while having a constraint on $D_{\ell d}(X,X_{t-1})$, which ensures that $X_t^{-1} - X_{t-1}^{-1}$ is not large.

---

### Author Response · Authors · 2022-11-18
**Improved readability**


We’ve made some changes in the revision to improve readability of the paper. Here’s a brief summary of the major changes:

* In Section 3.2, we presented the algorithm early in the section and then explained in two parts **Maintaining $H_t\in S_n(\mathcal{G})$ in line 4** and **Computing $X_t$ in line 5**.
 * Explicit solution to (11) for chain graph is mentioned separately from banded in Theorem 1.
 * In addition to some of the errors pointed out by reviewers, we fixed the following errors:
     * The second term in (3) is changed to  $(w_{t} - w^*)^T(X_{t}^{-1}-X_{t-1}^{-1})(w_{t} - w^*)$ in the paper, since $X_{t}^{-1}-X_{t-1}^{-1}$ might not be positive definite.
     * $H_t \in S_n(\mathcal{G})^{++}$ is changed to $H_t \in S_n(\mathcal{G})$ throughout the paper, since, $H_t$ as generated by line 4 in Algorithm 1 need not be positive definite.

---

### Author Response · Authors · 2022-11-22
**Reminder about Rebuttal**

Gentle reminder that we have submitted our rebuttal to each reviewer after conducting more experiments and editing paper accordingly.

---

### Decision · Program_Chairs · 2023-01-20

**Decision:**

Reject

**Justification For Why Not Higher Score:**

All reviewers unanimously agree the current version of the manuscript is not ready for publication (below the accept line), so is the decision.

**Justification For Why Not Lower Score:**

N/A

**Metareview: Summary, Strengths And Weaknesses:**

This manuscript proposed SONew (Sparsified Online Newton), an online convex optimizer derived by combining the regret minimization based on the LogDet divergence measure and structured sparsity in the preconditioning matrix. The preconditioning matrix of the SONew is a shape of either diagonal, tri-diagonal, or banded, which is an efficient variant of online Newton step (ONS) by using b-banded sparsity graph. On the MNIST autoencoder training task, SONew achieved better convergence than the existing first-order adaptive gradient method. Furthermore, SONew shows better numerical stability than other methods, including the Shampoo optimizer, when training with bfloat16. Main concerns of reviewers are about: 1) theoretical justification of the algorithm may have some issues, e.g. in regret bound; 2) experimental evaluation is insufficient, especially comparisons with relevant literature etc. All reviewers unanimously agree the current version of the manuscript is not ready for publication, so is the decision.